# Polarity signaling ensures epidermal homeostasis by coupling cellular mechanics and genomic integrity

Martim Dias Gomes [1,3], Soriba Letzian [1,3], Michael Saynisch [1] & Sandra Iden [1,2]

Epithelial homeostasis requires balanced progenitor cell proliferation and differentiation, whereas disrupting this equilibrium fosters degeneration or cancer. Here we studied how cell polarity signaling orchestrates epidermal self-renewal and differentiation. Using genetic ablation, quantitative imaging, mechanochemical reconstitution and atomic force microscopy, we find that mammalian Par3 couples genome integrity and epidermal fate through shaping keratinocyte mechanics, rather than mitotic spindle orientation. Par3 inactivation impairs RhoA activity, actomyosin contractility and viscoelasticity, eliciting mitotic failures that trigger aneuploidy, mitosis-dependent DNA damage responses, p53 stabilization and premature differentiation. Importantly, reconstituting myosin activity is sufficient to restore mitotic fidelity, genome integrity, and balanced differentiation and stratification. Collectively, this study deciphers a mechanical signaling network in which Par3 acts upstream of Rho/actomyosin contractility to promote intrinsic force generation, thereby maintaining mitotic accuracy and cellular fitness at the genomic level. Disturbing this network may compromise not only epidermal homeostasis but potentially also that of other self-renewing epithelia.

---

[1] Cologne Excellence Cluster on Cellular Stress Responses in Aging-Associated Diseases (CECAD), University of Cologne, Cologne 50931, Germany. [2] Center for Molecular Medicine Cologne (CMMC), University of Cologne, Cologne 50931, Germany. [3]These authors contributed equally: Martim Dias Gomes, Soriba Letzian. Correspondence and requests for materials should be addressed to S.I. (email: sandra.iden@uk-koeln.de)

Epithelial sheets form important dynamic barriers between the interior of an organism and its environment, provide mechanical protection or mediate secretion, absorption, and sensory functions. Being of central importance to survival, mechanisms have evolved to ensure that epithelial integrity is maintained during growth and quickly re-established upon injury. The epidermis is the major functional unit of the mammalian skin barrier and protects against harmful external insults or uncontrolled water loss[1]. In the developing epidermis, spindle orientation is coordinated with the specific need to produce either basal layer, self-renewing progeny, or keratinocytes that commit to terminal differentiation and move suprabasally[2–4]. Whereas postnatal loss of spindle orientation alters the fate of hair matrix cells during hair follicle morphogenesis[5], evidence emerges that in the adult epidermis also alternative mechanisms control cell fate decisions independent of spindle orientation. During homeostasis of adult ear skin, adjacent cells have recently been shown to undergo coupled, opposite fate decisions, with self-renewal being triggered by neighbor differentiation[6]. Moreover, accumulation of DNA damage in adult mice has been implicated in HFSC loss and increased epidermal differentiation[7], indicating that genome instability may serve as another parameter mediating tissue degeneration.

The function of many tissues also depends on a tight spatio-temporal control of cell and tissue architecture. Cell polarity refers to the unequal distribution of membrane compartments, organelles and/or macromolecules within a cell, and is critical for development, tissue homeostasis, and repair[8,9]. Work in invertebrates identified the Par complex, consisting of Partitioning-defective3 (Par3), atypical protein kinase C (aPKC), and Par6, as a key module driving polarity processes including apico-basal polarity, asymmetric cell division and spindle orientation[4,10]. Recent work including ours showed that proteins of the Par3/aPKC complex control adult hair follicle stem cell (HFSC) maintenance and differentiation[11–13]. Altered spindle orientation was unlikely to cause these defects, since Par3-deficient epidermis displayed a shift to more planar and oblique cell divisions[12] instead of increased perpendicular divisions that are thought to fuel differentiation[2–4]. Therefore, we sought to investigate alternative molecular mechanisms through which polarity proteins mediate stem cell maintenance and control epidermal differentiation.

Here, we provide evidence that Par3 controls epidermal integrity and fate predominantly through shaping Rho-dependent keratinocyte mechanics. By combining biophysical approaches including atomic force spectroscopy with mouse genetics, pharmacological treatments, and quantitative time-lapse imaging, our loss-of-function and gain-of-function approaches uncover that polarity protein-dependent actomyosin contractility is essential to safeguard genome integrity, thereby balancing fate decisions in the mammalian epidermis. We show that loss of the polarity protein Par3 reduces RhoA and myosin activity, leading to lower cortical tension, which predisposes keratinocytes to mitotic failures, aneuploidy, and progressive accumulation of DNA damage. The resulting genome instability and DNA damage responses subsequently fuel premature epidermal differentiation, likely involving p53. This study thus identifies mammalian Par3 as a key integrator of Rho-driven keratinocyte mechanics. Moreover, it presents a polarity protein-mediated mechanism that governs tissue homeostasis through coupling mechanical forces with mitotic accuracy and genome integrity important to balance self-renewal and differentiation in a barrier-forming epithelium.

## Results

### Epidermal Par3 loss leads to DNA damage responses. In search of potential mechanisms how epidermal deletion of *Par3* leads to

progressive decline of HFSCs and/or ectopic epidermal differentiation[12], we analyzed murine skin at postnatal day 100 (P100), a time-point right before Par3-deficient HFSCs start to decline[12]. Immunofluorescence analyses revealed increased signs of DNA damage throughout the skin epithelium of epidermal *Par3*KO (*Par3*eKO) mice, with γH2Ax-positive cells located in the interfollicular epidermis (Fig. 1a, f) and in various compartments of the hair follicle, including the bulge (Fig. 1b–g, Supplementary Fig. 1a, b). In line with increased numbers of γH2Ax-positive cells, Par3 loss in keratinocytes resulted in a significant upregulation of the tumor suppressor p53 (Fig. 1h, i, Supplementary Fig. 1c), indicating genomic stress. We next assessed the activation of DNA damage checkpoint kinases earlier shown to activate and stabilize p53[14]. Intriguingly, loss of Par3—but not K14-promoter-driven Cre expression alone—resulted in increased basal activity of the upstream checkpoint kinase ATR, and ectopic activation of both ATR and Chk1 kinases following irradiation with UV-B, a physiological stressor of epidermal cells (Fig. 1j, k, Supplementary Fig. 1d). Together, these data indicate that disturbed polarity protein function mounts ectopic DNA damage responses in the skin epithelium and epidermal keratinocytes.

### Par3 ensures mitotic fidelity and genome integrity. We next sought to identify the potential causes of elevated DNA damage signaling. Although Par3 has previously been implicated in repair of γ-irradiation-induced DNA double-strand breaks[15], we did not obtain evidence for compromised DNA repair in Par3-deficient keratinocytes (Supplementary Fig. 1e). Considering our earlier observations of altered spindle orientation in *Par3*eKO mice[12] and the established link between aberrant chromosome segregation and DNA damage[16,17], we next investigated the consequence of Par3 loss for cell division. Interestingly, time-lapse microscopy of H2B-GFP-expressing keratinocytes revealed a significant increase of aberrant cell divisions in Par3-deficient cultures that included lagging strands, multipolar divisions, and cytokinesis failure (Fig. 2a, b), accompanied by shortened mitotic duration (Fig. 2c). Moreover, the mitotic defects correlated with a significant increase of aneuploidy in Par3-deficient keratinocytes as determined by interphase fluorescence in situ hybridization (iFISH) targeting chromosome 2 (Fig. 2d, e), providing a potential explanation for the observed p53 induction[18].

Above findings unveiled an important role of Par3 for proper execution of mitosis and for maintaining genome integrity in mammalian epithelial cells. To directly assess if the mitotic defects were related to the elevated DNA damage responses observed in the absence of Par3 we blocked mitosis using the Cdk1 inhibitors Purvalanol-A or RO3306. Strikingly, mitotic block was able to significantly reduce the increased ATR activity in Par3-deficient cells (Fig. 2f, g, Supplementary Fig. 2a, b), supporting our hypothesis that erroneous mitosis following Par3 loss contributes to elevated DNA damage signals. Of note, compared to vehicle-treated cells, Purvalanol-A also led to a slight though non-significant reduction of ATR phosphorylation in control cells. This was potentially due to the baseline, Par3-independent mitotic inaccuracy of primary keratinocytes (Fig. 2b), a phenomenon that was recently reported for different primary cell culture systems[19].

### Par3 promotes cell dynamics, RhoA, and myosin activity. Having established that Par3 inactivation elicits disturbed mitosis and subsequent accumulation of DNA damage, we next aimed to understand the upstream events responsible for mitotic infidelity. First, the role of Par3 for overall cellular dynamics during cell division was examined employing detailed time-lapse microscopy

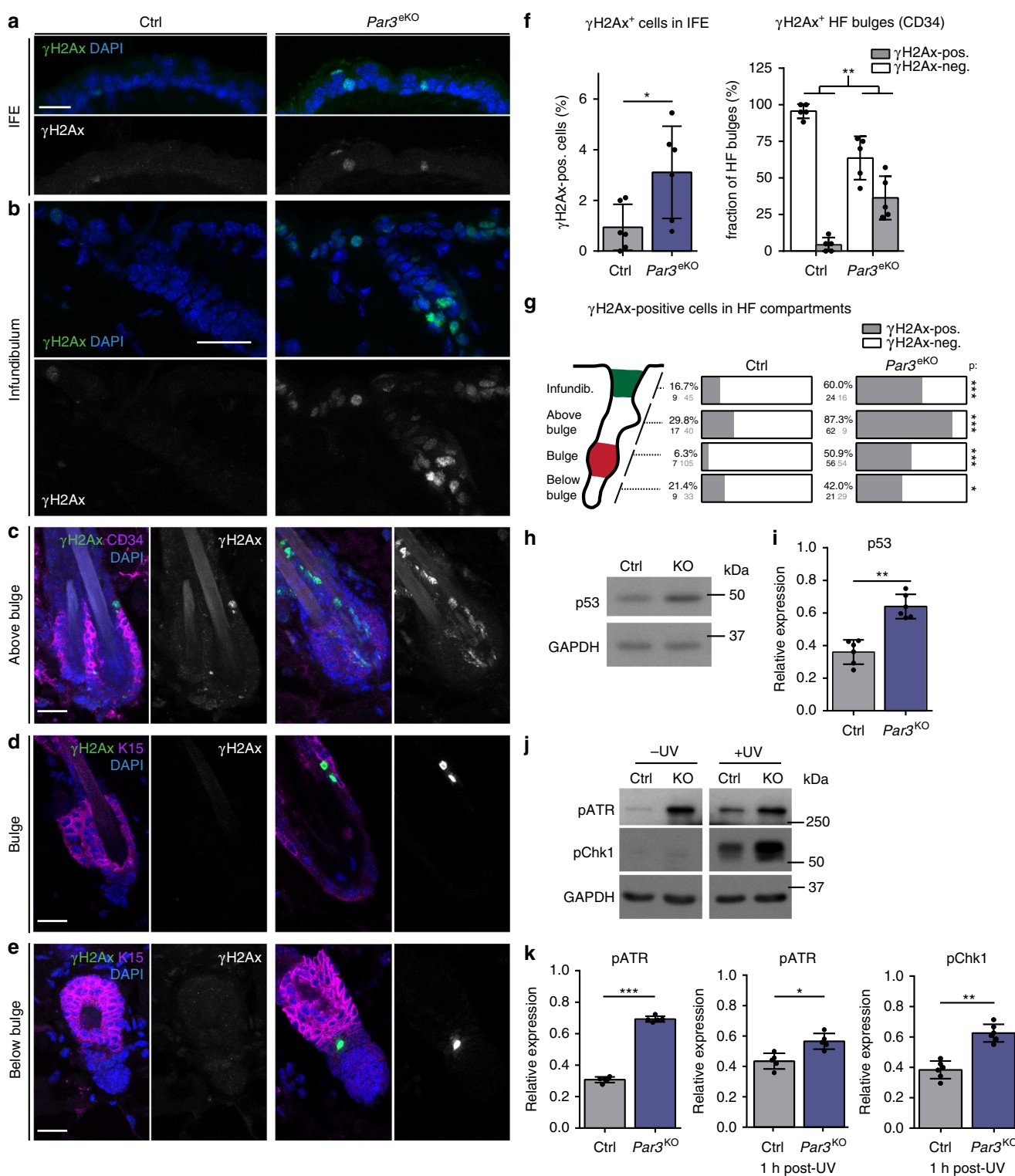

of H2B-GFP-labeled keratinocytes followed by particle image velocimetry (PIV)[20]. This analysis revealed a significantly reduced strain rate of mitotic Par3-deficient keratinocytes as compared to control cells (Fig. 3a, b), suggesting that Par3 orchestrates keratinocyte cell mechanics. During mitosis, cells undergo significant changes in shape, mechanics and polarity[21,22] important for equal distribution of the genetic material[23,24]. Mitotic rounding, a RhoA-driven actomyosin-dependent process[24], is important for bipolar spindle formation[25]. Par3 orthologs in *C. elegans* and *Drosophila* have previously been linked to actomyosin

contractility[26–29], albeit that the specific hierarchy among myosin activation and Par3 appears to be context-dependent. Based on the junctional localization of Par3 in keratinocytes[30] (Supplementary Fig. 3a) we hypothesized that the mitotic aberrations could be caused by a primary defect in generating spatiotemporal contractile stresses at cell–cell contacts. Immunofluorescence analysis of primary keratinocytes indeed revealed a significant decrease of phosphorylated myosin light chain 2 (pMLC2) at intercellular adhesions of *Par3*KO keratinocytes (Fig. 3c, d), which could be recapitulated by siRNA-mediated depletion of

**Fig. 1** Epidermal Par3 loss leads to increased DNA damage and ectopic activation of DNA damage responses. **a–e** Maximum projection of back-skin cross sections from P100 old epidermal *Par3*A knockout (*Par3*eKO: K14Cre/+;*Par3*fl/fl) and control mice (K14Cre/+) stained for γH2Ax and Keratin15 or CD34. Micrographs show (**a**) the interfollicular epidermis (IFE), Scale bar: 15 μm (**b**) infundibulum, (**c**) isthmus and junctional zone (above bulge), (**d**) bulge, (**e**) secondary hair germ, bulb, and dermal papilla (below bulge). Scale bars: 20 μm. **f** Quantification of γH2Ax-positive cells in the IFE, and percentage of HF bulges (labeled by CD34) with cells positive for γH2Ax. n(IFE) = 6 mice, *p = 0.0250, mean ± SD, paired two-tailed Student's t-test; n(HF) = 5 mice; **p = 0.0018; mean ± SD, two-way ANOVA/Sidak's multiple comparison). **g** Quantification and graphical illustration of γH2Ax-positive cells in different HF compartments that were distinguished based on bulge marker expression and histological hallmarks (as shown in **a–e**): Data represent pooled numbers of positive cells from five individual animals per genotype. Total numbers of γH2Ax-positive (dark gray) and γH2Ax-negative compartments (light gray) per genotype are shown. *p = 0.0455, ***p < 0.0001; two-sided Fisher's exact test (2 × 2 contingency table). **h** Immunoblot for p53 in whole cell keratinocyte lysates. GAPDH served as loading control. **i** Quantification of **h**. p53 levels were normalized to GAPDH and then expressed as relative values. n = 6 independent experiments, paired two-tailed Student's t-test, **p = 0.0058, mean ± SD. **j** Immunoblot analysis of pATR and pChk1 in keratinocytes either non-treated or UV-B-treated (100 mJ/cm$^2$). GAPDH served as loading control. **k** Quantification of **j**. pATR levels were normalized to GAPDH and then expressed as relative values. pChk1 levels were normalized to Chk1 and then expressed as relative values. n(pATR no UV, 1 h post-UV) = 5 independent experiments; n(pChk1 1 h post-UV) = 6 independent experiments, paired two-tailed Student's t-test, *p = 0.0478, **p = 0.0038, ***p < 0.0001; mean ± SD. Cropped immunoblot data are shown. Image intensity was enhanced for better visualization. HF hair follicle, Ctrl control, KO *Par3*KO, IFE interfollicular epidermis, infund. infundibulum, neg negative, pos positive

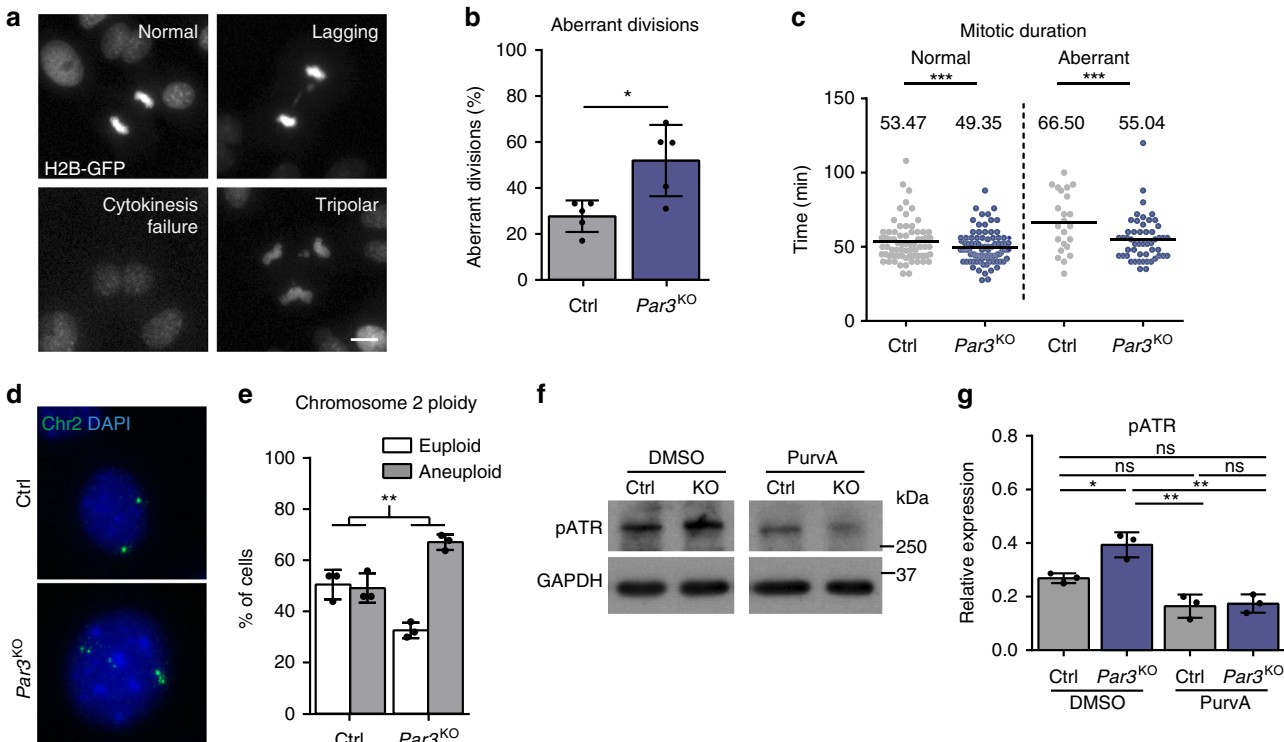

**Fig. 2** Par3 ensures mitotic fidelity and genome integrity of keratinocytes. **a** Representative images of mitotic aberrations observed in time-lapse microscopy of H2B-GFP-expressing keratinocytes (H2B-GFP in gray). Scale bar: 10 μm. **b** Quantification of aberrant cell divisions in primary *Par3*KO and control keratinocytes observed in time-lapse microscopy. n = 5 independent experiments, paired two-tailed Student's t-test, *p = 0.0438, mean ± SD. **c** Quantification of mitotic duration (Ctrl/normal n = 69 cells, *Par3*KO/normal n = 95 cells, Ctrl/aberrant n = 23 cells, *Par3*KO/abnormal n = 51 cells; in all cases pooled from three independent experiments). For statistical analysis a nonlinear mixed model using RStudio software was employed, yielding ***p < 0.001. Bar represents mean. **d** Representative images of iFISH probes targeting chromosome 2. Scale bar: 10 μm. **e** Quantification of **d**. n = 3 independent experiments, **p = 0.0028, mean ± SD; two-way ANOVA/Sidak's multiple comparisons test. **f** Immunoblot analysis of ATR activation in primary Par3-deficient and control keratinocytes after treatment with Cdk1 inhibitor Purvalanol A (10 μM). GAPDH served as loading control. **g** Quantification of **f**. pATR levels were first normalized to GAPDH and then expressed as relative values. n = 3 independent experiments; mean ± SD; two-way ANOVA/Tukey's multiple comparisons test *p = 0.0459, **p = 0.0026 (*Par3*KO/DMSO vs. Ctrl/PurvA), **p = 0.0032 (*Par3*KO/DMSO vs. *Par3*KO/PurvA). Cropped immunoblot data are shown. iFISH interphase fluorescence in situ hybridization, Ctrl control, KO *Par3*KO, PurvA Purvalanol A, ns non-significant

Par3 in wild-type keratinocytes (Fig. 3e–g). In agreement, immunoblotting demonstrated a significant overall reduction in phosphorylation of MLC2 and of the actin-binding proteins ezrin–radixin–moesin (pERM) (Fig. 3h, i). Moreover, both re-expression of Par3-full length (Par3-FL) and membrane-targeted Par3 (Par3-CAAX) were sufficient to restore junctional MLC2 phosphorylation in *Par3*KO keratinocytes (Supplementary

Fig. 4a, b), supporting a role for cortical Par3 upstream of myosin activation. Importantly, active RhoA (RhoA-GTP), but not total RhoA, was significantly decreased following Par3 inactivation (Fig. 3j, Supplementary Fig. 6a, b). This was further associated with increased protein expression of the Rho-inhibiting protein p190-RhoGAP (Fig. 3k, l). Consistent with impaired Rho/myosin activity in primary *Par3*KO keratinocytes,

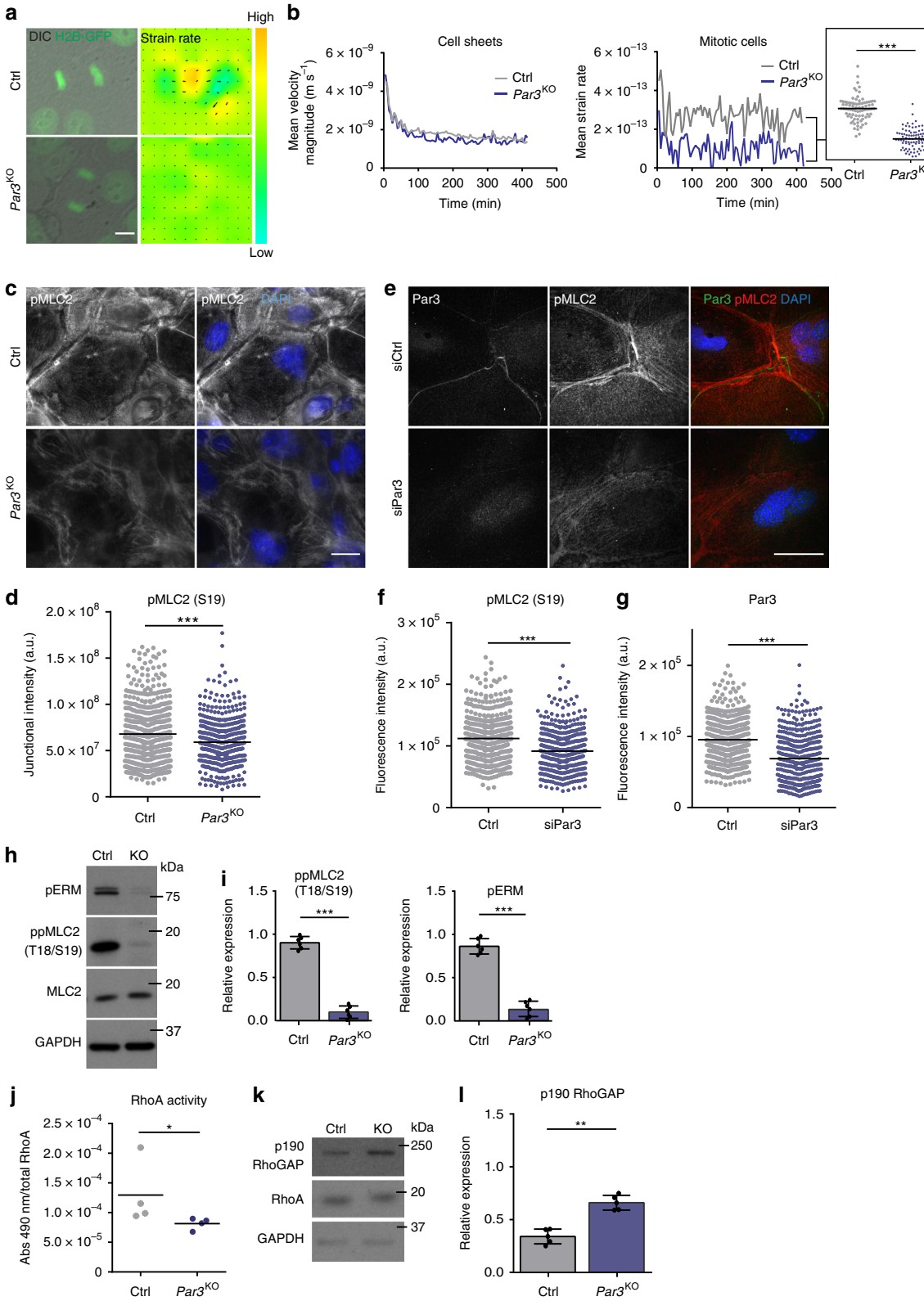

immunohistochemistry analyses of murine skin demonstrated significantly reduced phosphorylation of MLC2 (Fig. 4a–d; Supplementary Fig. 5a–g), as well as a trend to decreased ERM phosphorylation in adult Par3-deficient epidermis (P100) (Fig. 4e, f). Together, these results suggest that Par3 is important to sustain mechanochemical signaling in epidermal keratinocytes in vitro and in vivo.

**Myosin activation rescues force generation and elasticity.** We next sought to examine the significance of reduced Rho and myosin activity for the mechanical defects observed upon Par3 loss. To restore contractility in *Par3*KO cells, either a Rho-activating peptide (CN03, stabilizing endogenous RhoA-GTP) or a low dose of Calyculin-A, a phosphatase inhibitor and enhancer of myosin activity, were used. To functionally assess the

**Fig. 3** Par3 safeguards keratinocyte dynamics, and promotes RhoA activity and actomyosin contractility. **a** Snapshots from live cell-imaging videos monitoring H2B-GFP (green) fluorescence and DIC, smoothened strain rate maps from particle image velocimetry analyses (physics look-up table) and pseudo-color scale. Scale bar: 10 μm. **b** Quantification of mean velocity magnitude (n = 3 independent experiments) and mean strain rate over time (n = 30 mitotic cells pooled from three experiments). Dot plot (right panel) shows overall mean strain rate, Mann–Whitney U-test, ***p < 0.0001 (mitotic cells), mean. **c** Immunofluorescence micrographs of MLC2 phosphorylation (Ser19) (gray) in primary murine keratinocytes. DAPI is shown in blue. Scale bar: 25 μm. **d** Quantification of pMLC2 (Ser19) immunoreactivity at cell–cell junctions, intensity in arbitrary units; n(Ctrl) = 526 cells, n(Par3KO) = 519 cells pooled from three independent experiments; ***p = 0.0002, two-sided Mann–Whitney U-test, bar represents mean. **e** Immunofluorescence micrographs of control keratinocytes transfected with siCtrl or siPar3 and stained for Par3 (green) and pMLC2 (red). DAPI is shown in blue. Individual channels are displayed in gray. Scale bar: 40 μm. **f**, **g** Quantification of pMLC2 **f** and Par3 **g** immunoreactivity at cell–cell junctions, intensity in arbitrary units; n = 450 cells pooled from five independent experiments; ***p < 0.0001, two-sided Mann–Whitney U-test, bar represents mean. **h** Immunoblot analysis of ppMLC2 (Thr18/Ser19) and pERM in whole cell keratinocyte lysates. Total MLC2 and GAPDH served as loading control, respectively. **i** Quantification of **h**. ppMLC2 (Thr18/Ser19) levels were normalized to total MLC and then expressed as relative values. n = 6 biologically independent samples, paired two-tailed Student's t-test, ***p < 0.0001, mean ± SD. pERM levels were first normalized to GAPDH and then expressed as relative values. n = 6 biologically independent samples, paired two-tailed Student's t-test, ***p = 0.0002, mean ± SD. **j** Quantification of RhoA G-LISA® activation assay from keratinocyte lysates. Absorbance was normalized to total RhoA level determined by western blot. n = 4 biologically independent samples, Mann–Whitney U-test, *p = 0.0286, mean. **k** Immunoblot analysis of p190-RhoGAP. GAPDH was used as loading control. **l** Quantification of **k**, p190-RhoGAP levels were first normalized to RhoA and then expressed as relative values. n = 5 biologically independent samples, paired two-tailed Student's t-test, **p = 0.0068, mean ± SD. Cropped immunoblot data are shown. Ctrl control, KO Par3KO, abs absorbance, DIC differential interference contrast

role of Par3 in intrinsic force generation, we performed epithelial sheet contraction assays (Fig. 5a). After Dispase-mediated lift, Par3-deficient epithelial sheets showed significantly impaired contraction when compared to control cell sheets (Fig. 5b–d), indicative of reduced internal stress. Strikingly, contraction of Par3KO keratinocyte sheets could be rescued either by treatment with Calyculin-A (Fig. 5b, c; Supplementary Fig. 6c, d) or the Rho-activating peptide (Fig. 5d), indicating that Par3 acts upstream of Rho/myosin to mediate myosin motor activity and intrinsic contractile forces.

As the actomyosin network is a major determinant of mechanical properties of cells and their responses to stress, we next employed atomic force spectroscopy to quantitatively assess the consequence of Par3 loss for keratinocyte mechanics. Interestingly, nano-indentation experiments using either sharp or spherical cantilevers uncovered impaired viscoelastic properties upon Par3 inactivation, with a significantly reduced elastic modulus measured in Par3-deficient keratinocytes compared to controls (Fig. 5e, Supplementary Fig. 6e–g). Similar to epithelial sheet contraction, Calyculin-A treatment of Par3KO keratinocytes was sufficient to rescue the elastic modulus (Fig. 5e–g, Supplementary Fig. 6f, g), thus demonstrating that Par3 regulates viscoelastic properties through control of actomyosin-dependent contractile behavior in epidermal keratinocytes.

**Contractility-dependent mitotic infidelity and p53 response.** Next to their localization at keratinocyte junctions (Supplementary Fig. 3a), we noted a partial colocalization of Par3 and phosphorylated MLC2 during mitosis, i.e. at the prometaphase cortex and the cleavage furrow (Supplementary Fig. 7a). Moreover, in line with our observation in interphase cells and in the skin in vivo, both pMLC2 and pERM levels were reduced in mitotic Par3KO keratinocytes (Supplementary Fig. 7a–c), further corroborating that Par3 promotes the activation of these molecules previously implicated in cortical mechanics[31–34].

To elucidate how impaired actomyosin contractility in the absence of Par3 contributes to mitotic infidelity, we first characterized mitotic spindles in terms of their geometry. In metaphase control keratinocytes the two poles of a spindle exhibited the expected perpendicular orientation toward its metaphase plate (Fig. 6a). Intriguingly, however, Par3-deficient cells frequently showed a deviation of spindle angles relative to the equatorial chromosome plate, reflected by a significantly reduced coherency index in these cells (Fig. 6a–c). This altered spindle geometry was further associated with an increased spindle circularity in Par3KO

keratinocytes (Supplementary Fig. 7d). Strikingly, both types of metaphase spindle shape alterations could be rescued by Calyculin-A treatment (Fig. 6a, c, Supplementary Fig. 7d). Time-lapse microscopy combined with either CN03 or Calyculin-A treatment confirmed this was relevant for the outcome of mitosis as reconstitution of Rho and myosin activity in Par3-deficient keratinocytes was sufficient to re-establish mitotic accuracy (Fig. 6d, g; Supplementary Movies 1–6). In accordance with the observed mitosis-dependent DNA damage responses, both CN03 and Calyculin-A treatment also significantly suppressed the p53 upregulation in Par3-deficient keratinocytes to the levels of control cells (Fig. 6e, f, h, i). We note that increasing actomyosin contractility with either compound also decreased the p53 levels of control cells (Fig. 6e, f, h, i). This suggests that the baseline mitotic infidelity and chromosomal instability of primary keratinocytes (Fig. 2b, e), and potentially of other primary cell systems[19], to some extent involves mechanical imbalance. Together, these data unravel that insufficient contractility as consequence of Par3 inactivation leads to erroneous cell divisions and subsequent mounting of ectopic DNA damage responses.

**Low contractility upon Par3 loss drives differentiated fate.** Evidence emerges connecting DNA damage and aneuploidy with reduced self-renewal capacity and increased differentiation in flies[35] and mammals[36,37]. Recent studies in mouse skin and cultured keratinocytes associated DNA damage accumulation with keratinocyte differentiation[7,38]. Having demonstrated that Rho or myosin activation as well as inhibition of mitosis prevents ectopic DNA damage responses in the absence of Par3, we hypothesized that the reduced genome integrity due to insufficient force generation fuels differentiation upon Par3 loss. To directly test this, we assessed the impact of restored contractility on expression of the epidermal differentiation markers Keratin1 and Involucrin following $Ca^{2+}$-induced keratinocyte differentiation. Consistent with our previous findings of increased epidermal differentiation in Par3eKO mice[12], immunoblot analyses confirmed significant Keratin1 and Involucrin upregulation in primary Par3KO keratinocytes compared to controls (Fig. 7a, b, Supplementary Fig. 8a). Strikingly, Calyculin-A treatment was sufficient to reverse this effect (Fig. 7a, b, Supplementary Fig. 8a), highlighting that impaired actomyosin contractility contributes to premature differentiation in the absence of Par3. Notably, in support of our hypothesis that elevated DNA damage responses upon Par3 ablation contribute to increased differentiated fate, treatment with ATM and ATR inhibitors (Supplementary Fig. 8b)

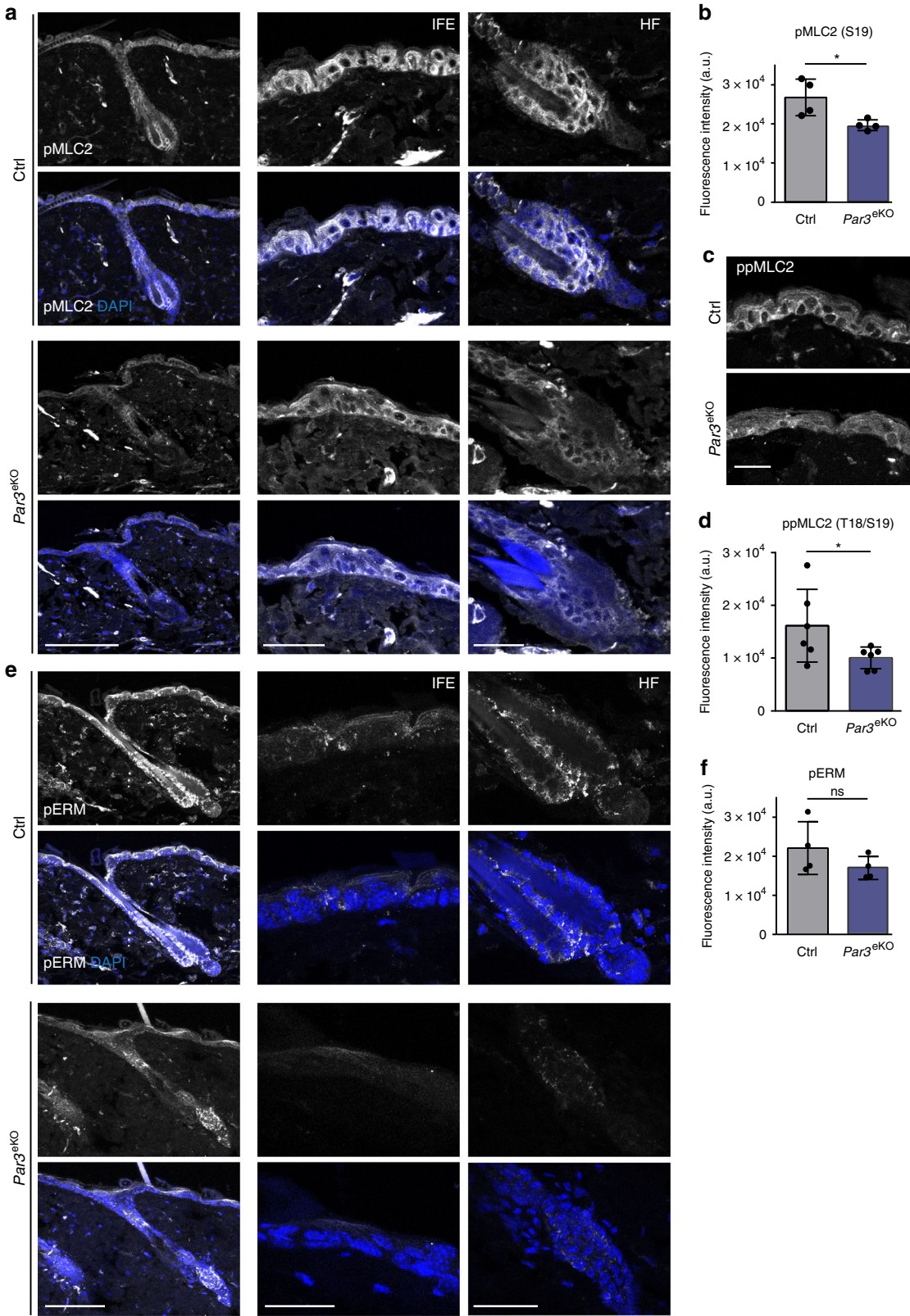

or siRNA-mediated depletion of p53 also resulted in reduced Keratin1 expression (Fig. 7c, d). Using this knock-down approach, however, *Par3*KO cells retained significantly higher levels of p53 when compared to control cells (Fig. 7c), likely explaining the residual difference in Keratin1 observed between sip53-transfected control and *Par3*KO cells (Fig. 7d). Quantitative RT-PCR analysis of Par3-deficient primary keratinocytes and

adult epidermis (P100) further revealed alterations in the expression of multiple transcriptional targets recently linked to p53[39,40]. Whereas common apoptosis-related targets like Bax[41] and Noxa[42] were unchanged (Supplementary Fig. 9a, b), we detected a significant upregulation of Tgm2[43], a protein of the transglutaminase family implicated in cellular differentiation[44]. Moreover, Par3 loss resulted in significant downregulation of the

**Fig. 4** Reduced phosphorylation of MLC2 and ERM in *Par3*eKO mice. **a** Immunohistochemistry of P100 murine back-skin paraffin cross-sections stained for pMLC2 (Ser19) (gray). Left panels: overview micrographs (×20 objective, scale bar: 100 μm). Middle and right panel: higher magnification micrographs referring to signals within the IFE (middle panel) and HF (right panel). ×40 objective, scale bar: 40 μm (IFE), 20 μm (HF). **b** Quantification of pMLC immunoreactivity **a**. Intensity in arbitrary units. $n = 4$ mice, Mann–Whitney U-test, *$p = 0.0286$, mean ± SD. **c** Immunohistochemistry of P100 murine back-skin paraffin cross-sections stained for ppMLC2 (Thr18/Ser19) (gray) (Cell Signaling Technologies #3674). ×63 objective, scale bar: 20 μm. **d** Quantification of ppMLC immunoreactivity **c**. Intensity in arbitrary units, $n = 6$ mice, Mann–Whitney U-test, *$p = 0.0411$, mean ± SD. **e** Immunohistochemistry of P100 murine back-skin paraffin cross-sections stained for pERM (gray). Left panels: overview micrographs (×20 objective, scale bar: 100 μm). Middle and right panel: higher magnification micrographs referring to signals within the IFE (middle panel) and HF (right panel). ×63 objective, scale bar: 20 μm (IFE and HF). Image intensity was enhanced for better visualization (×63). **f** Quantification of pERM immunoreactivity **e**. Intensity in arbitrary units. $n = 4$ mice, Mann–Whitney U-test, mean ± SD. In all micrographs DAPI is shown in blue. Ctrl control, IFE interfollicular epidermis, HF hair follicle, ns non-significant

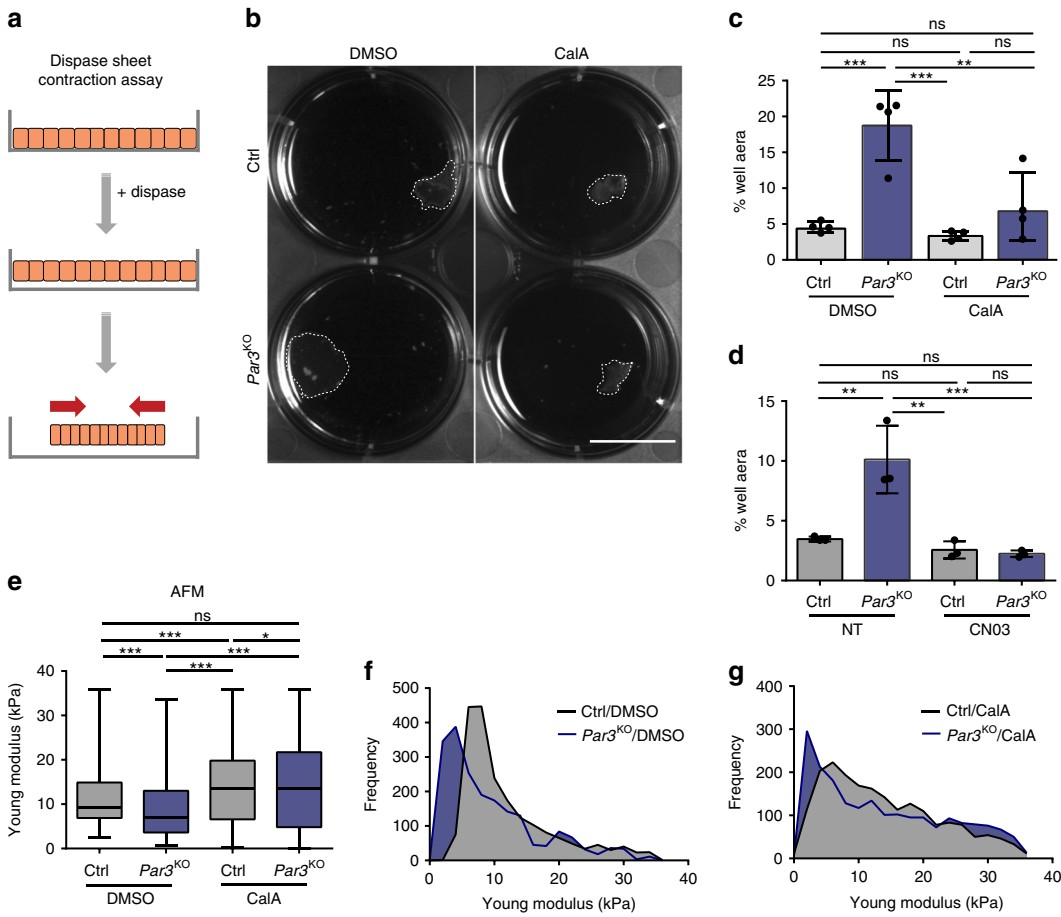

**Fig. 5** Re-establishment of actomyosin contractility rescues viscoelastic properties in Par3-deficient cells. **a** Schematic representation of dispase sheet contraction assay. **b** Representative images of dispase sheet contraction assay 24 h after 1 nM CalA treatment using control and *Par3*KO keratinocytes. Scale bar: 15 mm. **c** Quantification of keratinocyte sheet area after dispase sheet contraction assay 24 h following treatment with CalA (1 nM). $n = 4$ independent experiments; **$p = 0.0029$, ***$p = 0.0004$ (Ctrl/DMSO vs *Par3*KO/DMSO), ***$p = 0.0002$ (*Par3*KO/DMSO vs. Ctrl/CalA) mean ± SD; one-way ANOVA/Tukey's multiple comparisons test. **d** Quantification of keratinocyte sheet area after dispase sheet contraction assay 24 h following treatment with the Rho activator CN03 (5 μg/ml). $n = 3$ independent experiments; **$p = 0.0024$ (Ctrl/NT vs. *Par3*KO/NT), **$p = 0.0011$ (*Par3*KO/NT vs. Ctrl/CN03), ***$p = 0.0008$, one-way ANOVA/Tukey's multiple comparisons test. Mean ± SD. **e** Young Modulus box-plot based on force indentation spectroscopy ($n = 2000$ measurements on primary keratinocytes, pooled from three independent experiments; *$p = 0.0368$, ***$p < 0.0002$ (Ctrl/DMSO vs. Ctrl/CalA), ***$p < 0.0001$ (Ctrl/DMSO vs. *Par3*KO/DMSO), ***$p < 0.0001$ (*Par3*KO/DMSO vs. Ctrl/CalA), ***$p < 0.0001$ (*Par3*KO/DMSO vs. *Par3*KO/CalA), Kruskal–Wallis/Dunn's multiple comparison test, box plots show minimum (boundary of lower whisker), 25th percentile (lower boundary of box), median (center line), 75th percentile (upper boundary of box), and maximum (boundary of upper whisker). **f, g** Distribution histogram of Young Modulus upon DMSO **f** and 24 h CalA (1 nM) **g** treatment of experiments shown in **e**. Ctrl control, CalA Calyculin A, NT non-treated, ns non-significant

Wnt pathway component Disheveled 1 (Dvl1)[40], in line with a reported role for Wnt signaling in keeping stemness of skin stem/progenitor cells[45]. Though requiring further investigation, this data opened the possibility that increased epidermal differentiation due to Par3 loss at least in part involves gene expression changes downstream of p53.

To further corroborate the link between Par3, mechanical signaling and epidermal differentiation, we tested if the tight control of the actomyosin network was also crucial to direct keratinocyte fate toward upward movement into suprabasal layers. For this, primary keratinocytes were cultured for prolonged periods on porous membranes to enhance

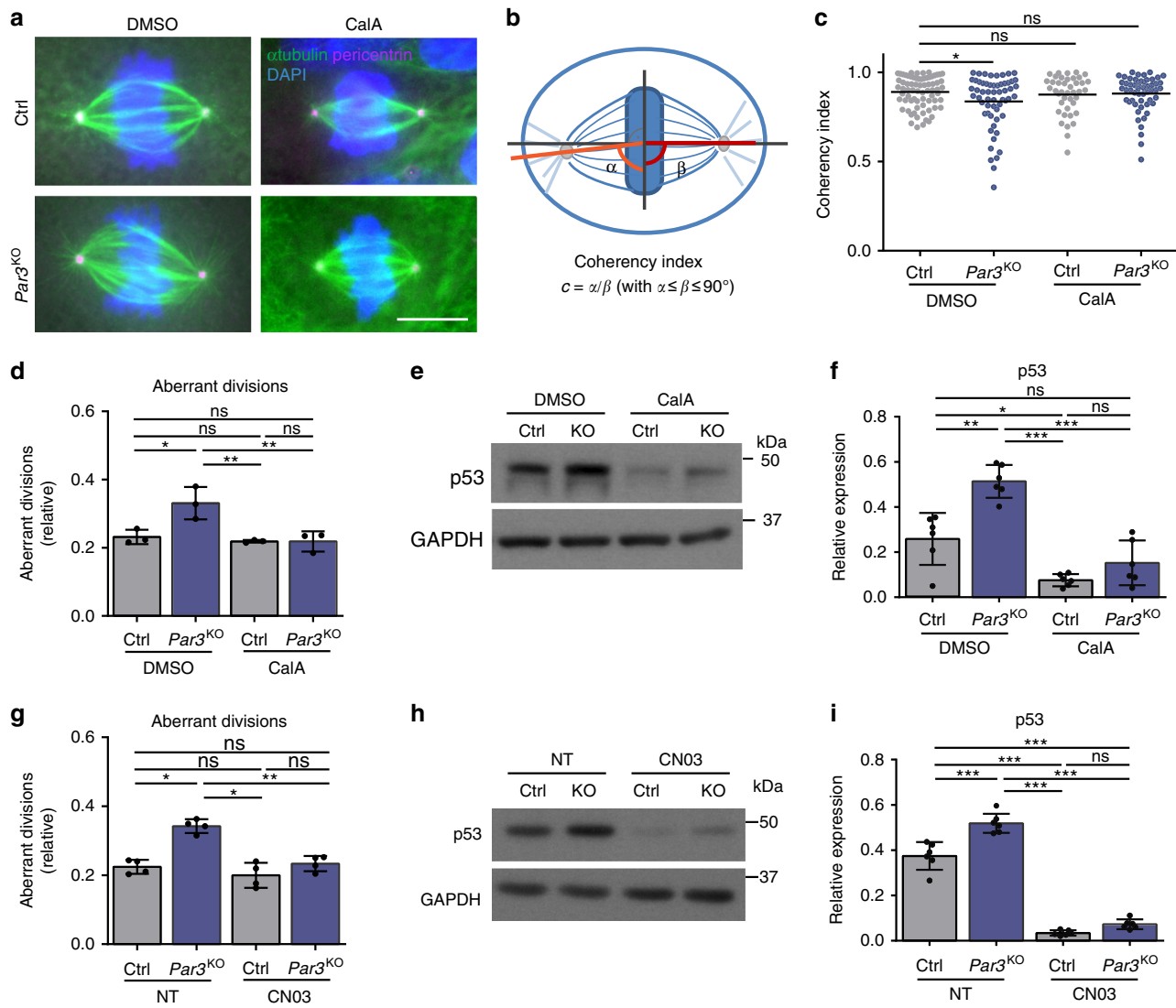

**Fig. 6** Restoration of Rho activity or actomyosin contractility rescues mitotic fidelity and p53 stabilization in *Par3*KO cells. **a** Immunofluorescence micrographs of mitotic keratinocytes stained for pericentrin (green) and alpha-tubulin (magenta), scale bar: 10 μm. Intensity was enhanced for better spindle visualization. **b** Schematic representation of analysis for mitotic spindle geometry. **c** Quantification of coherency index (Ctrl/DMSO-treated: $n = 73$ cells, *Par3*KO/DMSO-treated: $n = 58$ cells; Ctrl/CalA-treated: $n = 41$ cells, *Par3*KO/CalA-treated: $n = 55$ cells; in all cases pooled from four independent experiments), $*p = 0.0207$, bar represents mean, one-way ANOVA/Dunnett's multiple comparisons test. **d** Quantification of aberrant cell divisions in primary *Par3*KO and control keratinocytes observed in time-lapse microscopy upon DMSO and CalA (1 nM) treatment (relative values). $n = 3$ independent experiments, mean ± SD, $*p = 0.0154$ (Ctrl/DMSO vs. *Par3*KO/DMSO), $**p = 0.0078$ (*Par3*KO/DMSO vs. Ctrl/CalA), $**p = 0.0076$ (*Par3*KO/DMSO vs. *Par3*KO/CalA) one-way ANOVA/Tukey's comparison test. **e** Immunoblot analysis of p53 upon DMSO and CalA (1 nM) treatment. GAPDH served as loading control. **f** Quantification of **e**. p53 levels were first normalized to GAPDH and then expressed as relative values. $n = 6$ biological independent samples; $**p = 0.0022$ (DMSO-treated Ctrl vs. *Par3*KO); $***p < 0.0001$ (*Par3*KO DMSO-treated vs. CalA-treated); $***p < 0.0001$ (DMSO-treated *Par3*KO vs. CalA-treated Ctrl); $*p = 0.0261$ (Ctrl/DMSO vs. Ctrl/CalA-treated); mean ± SD; two-way Anova/Tukey's multiple comparison test. **g** Quantification of aberrant cell divisions in primary *Par3*KO and control keratinocytes observed in time-lapse microscopy either non-treated (NT) or treated with 0.5 μg/ml CN03 (relative values). $n = 4$ biological independent samples, mean ± SD, $*p = 0.0236$ (Ctrl/NT vs. *Par3*KO/NT), $*p = 0.0349$ (*Par3*KO/NT vs. Ctrl/CN03), $**p = 0.0016$ (*Par3*KO/NT vs. *Par3*KO/CN03), one-way ANOVA/Tukey's comparison test. **h** Immunoblot analysis of p53 from NT or CN03 treated primary *Par3*KO and control keratinocytes. GAPDH was used as loading control. **i** Quantification of **h**. p53 level were first normalized to GAPDH and then expressed as relative values. $n = 6$ biological independent samples; $***p = 0.0003$ (Ctrl/NT vs. *Par3*KO/NT), $***p < 0.0001$, mean ± SD; two-way ANOVA/Tukey's multiple comparison test. Cropped immunoblot data are shown. Ctrl control, KO *Par3*KO, CalA Calyculin A, NT non-treated, ns non-significant

keratinocyte stratification into basal and suprabasal keratinocyte layers, more closely mimicking the in vivo tissue architecture. Primary control and *Par3*KO keratinocytes were mixed and seeded at a 50/50% ratio, using H2B-GFP as genetic label distinguishing genotypes (Fig. 7e). After 10 days of culture stratification was achieved, and the number of control vs. *Par3*KO cells in basal and suprabasal layers was examined.

These experiments revealed a consistent enrichment of Par3-deficient cells in the suprabasal layer relative to control cells (Fig. 7e–g), thus illustrating that *Par3*KO keratinocytes were more likely to move upwards, in line with a predisposition to differentiation. Most importantly, restoring actomyosin contractility by low dose Calyculin-A could revert the ratio of suprabasal control vs. *Par3*KO cells (Fig. 7e–g), emphasizing

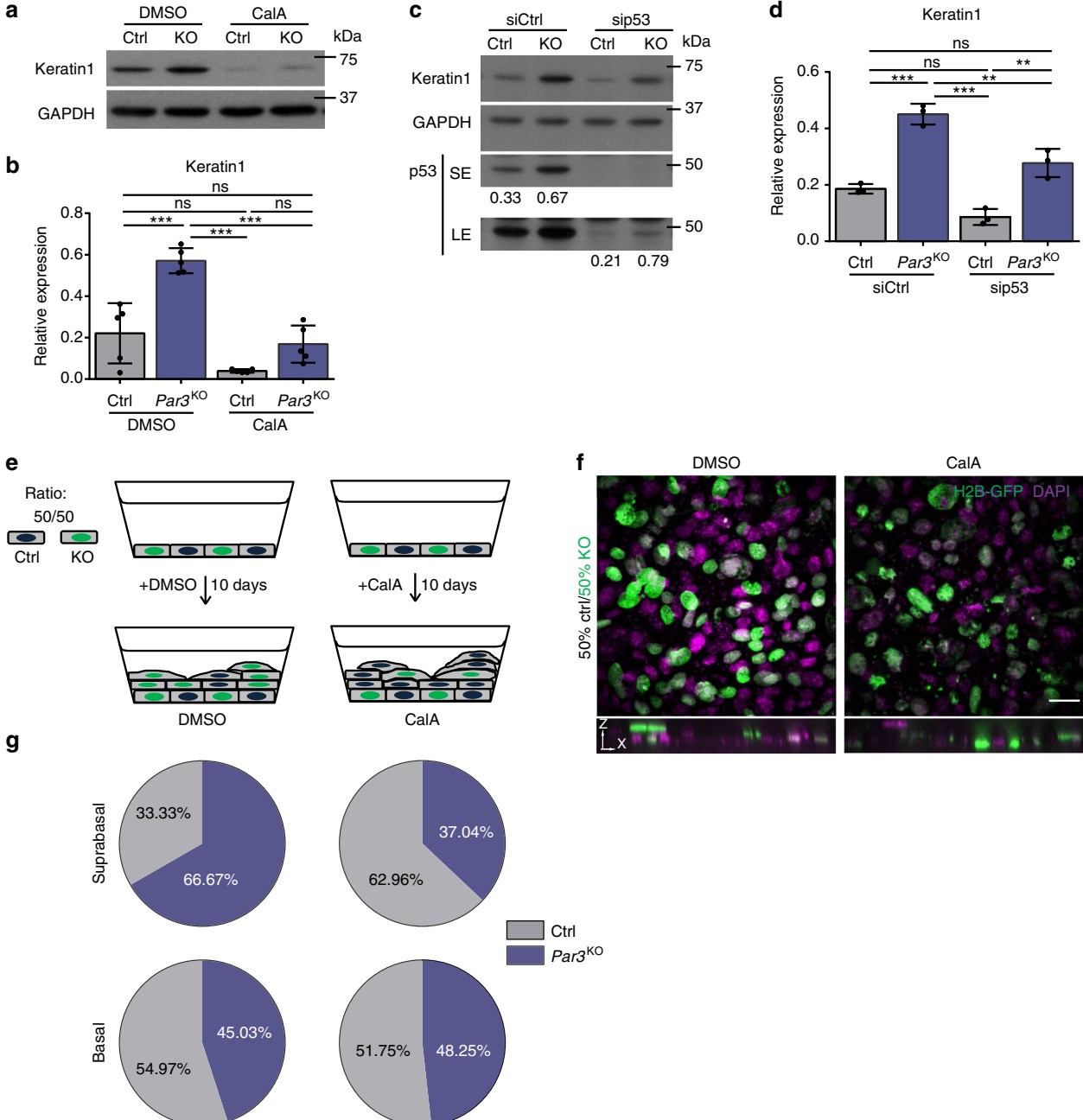

**Fig. 7** Reduced actomyosin contractility following Par3 loss drives ectopic differentiation and suprabasal fate. **a** Immunoblot analysis for Keratin1 expression in keratinocyte lysates following 48 h CalA (1 nM) treatment. GAPDH served as loading control. **b** Quantification of **a**. Keratin1 levels were first normalized to GAPDH and then expressed as relative values. $n = 5$ biological independent samples; ***$p = 0.001$ (Ctrl vs. *Par3*KO, DMSO-treated); ***$p = 0.0003$ (*Par3*KO, DMSO-treated vs. CalA-treated), ***$p < 0.0001$ (*Par3*KO/DMSO-treated vs. Ctrl/CalA-treated), mean ± SD, two-way ANOVA/Tukey's multiple comparisons test. **c** Immunoblot analysis for Keratin1 expression in keratinocyte lysates following siCtrl or sip53 transfection. GAPDH served as loading control. Relative p53 protein levels are shown below the immunoblot. **d** Quantification of **c**. Keratin1 levels were first normalized to GAPDH and then expressed as relative values. $n = 3$; ***$p = 0.0009$ (Ctrl vs. *Par3*KO, siCtrl-treated); ***$p = 0.0001$ (*Par3*KO siCtrl-treated vs. Ctrl sip53-treated); **$p = 0.008$ (*Par3*KO siCtrl-treated vs. sip53-treated); **$p = 0.0049$ (Ctrl sip53-treated vs. *Par3*KO sip53-treated); mean ± SD; two-way ANOVA/Tukey's multiple comparisons test. **e** Schematic representation of the Transwell filter culture system and experimental setup. *Par3*KO keratinocytes were positive for H2B-GFP, control keratinocytes isolated from littermates were negative for H2B-GFP. **f** Immunofluorescence micrographs of stratified cultures incubated for 10 days either in DMSO or in 1 nM CalA. *Par3*KO keratinocytes were H2B-GFP (green) labeled, DAPI is shown in magenta. Bottom panels show zx-projection. Scale bar: 50 μm. **g** Pie graphs showing cell distribution according to basal or suprabasal position following a 10-day culture period of mixed control and *Par3*KO keratinocyte cultures on Transwell filters. DMSO-treated samples: $n = 180$ cells (basal/DMSO-treated: $n = 171$ cells; suprabasal/DMSO-treated: $n = 9$ cells); CalA-treated samples: $n = 170$ cells (basal/CalA-treated: $n = 143$ cells; suprabasal/CalA-treated: $n = 27$ cells). An experimental representaton of three independent experiments is shown. Cropped immunoblot data are shown. Image intensity was enhanced for better visualization. Ctrl control, KO *Par3*KO, CalA Calyculin A, ns non-significant, SE short exposure, LE long exposure, siCtrl small interfering non-targeting RNAs, sip53 small interfering RNAs targeting p53

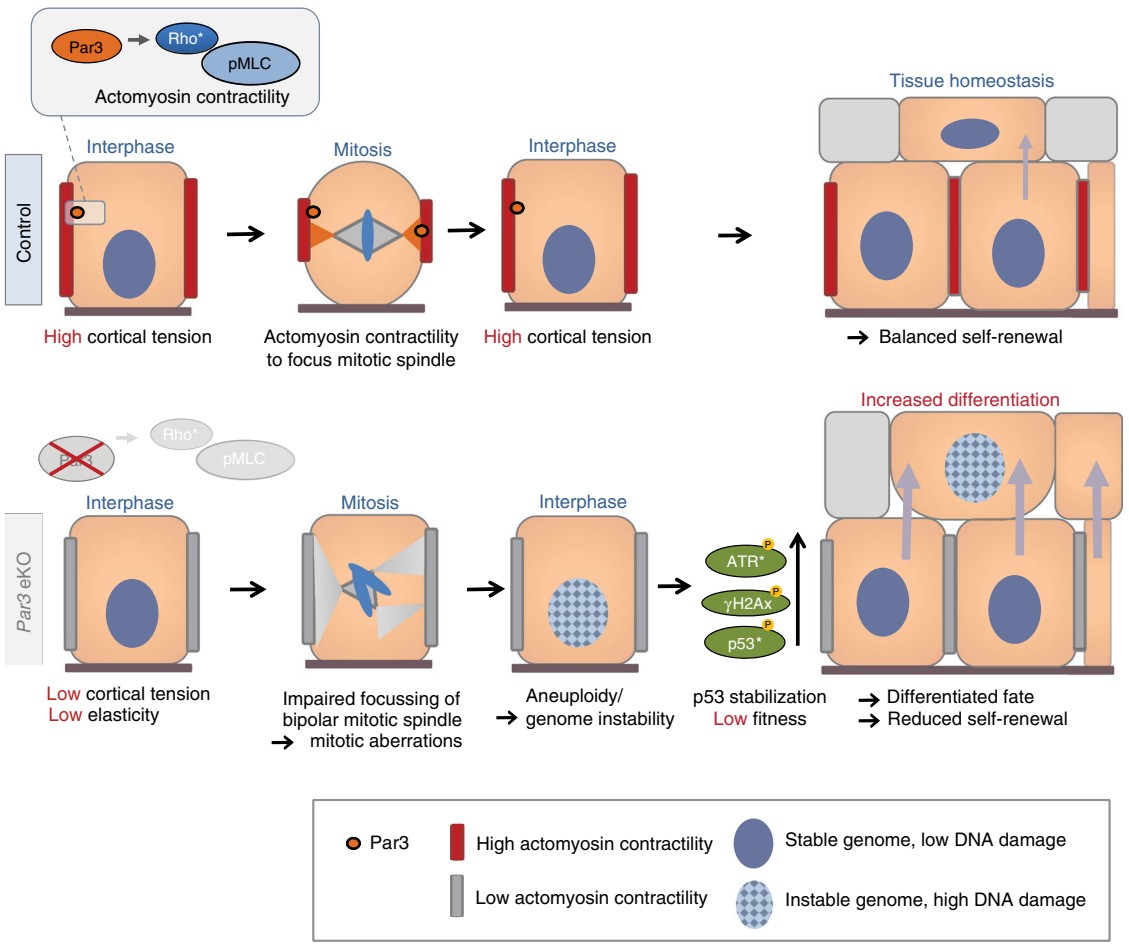

**Fig. 8** Par3 directs epidermal fate decisions through coupling Rho/actomyosin contractility to mitotic fidelity. Model. Par3 regulates actomyosin contractility in a Rho-dependent manner. Loss of Par3 alters keratinocyte mechanics resulting in mitotic infidelity and accumulation of DNA damage and p53, which in turn fuels differentiation and disturbs epithelial homeostasis. Restoring myosin activity in Par3-deficient keratinocytes is sufficient to normalize mitosis, and to prevent ectopic DNA damage, premature differentiation, and suprabasal fate. These data establish a role of Par3 in keratinocyte mechanics and mitotic fidelity. Rather than promoting asymmetric/perpendicular mitotic spindle orientation, altered polarity protein function in adult epithelia causes ectopic differentiated fate due to impaired cortical contractility that yields mitosis-related DNA damage and as consequence lower fitness

that low myosin activity in Par3-deficient cells is a crucial driver of suprabasal, differentiated fate.

## Discussion

In self-renewing tissues, cell divisions need to be tightly controlled to maintain a genetically and quantitatively stable stem/progenitor cell pool while producing sufficient numbers of daughter cells that commit to differentiation and serve overall tissue function. Our biomechanical, genetic, and imaging approaches uncover mechanisms through which the polarity protein Par3 instructs faithful mitosis and thereby ultimately epidermal fate decisions. We provide in vivo and in vitro evidence that Par3 acts upstream of Rho/actomyosin activation to control epidermal homeostasis, illuminating a distinct hierarchy of polarity and mechanical signaling in the skin. Based on our findings we propose a model (Fig. 8) in which junctional Par3-safeguards genome integrity through the control of actomyosin contractility and keratinocyte mechanics to ensure mitotic fidelity. Par3 inactivation leads to impaired RhoA and myosin activities, and causes mitotic infidelity, aneuploidy, and DNA damage responses yielding p53 stabilization. Mechanistically, we demonstrate that mitosis is required to mount the increased DNA damage responses in Par3-deficient keratinocytes, and that restoring myosin activity is sufficient to normalize mitosis and to

prevent ectopic DNA damage, premature differentiation, and suprabasal fate.

Our data further strongly suggest that genome instability and the resulting DNA damage responses act as defining factors for differentiated epidermal cell fate. These data are in agreement with recent developmental studies linking p53 to embryonic stem cell differentiation[46] and data from mouse embryos exposing p53 as driver of differentiated fate by reducing cell competition and fitness[47,48]. Our findings extend these developmental data to a more general concept that orchestrates tissue homeostasis. In light of the frequent extrinsic and intrinsic damages that cells in adult and aging organisms are facing over time, tissues may utilize biochemical alert signals reflecting compromised genome quality as a parameter for fate decisions to sustain long-term stem/progenitor cells with highest genome integrity, whereas cells exhibiting (moderate) DNA damage are eliminated by commitment to differentiation.

Although our findings clearly implicate defective mitosis in the increased epidermal differentiation, they also show that Par3 controls keratinocyte mechanics both during cell division and in interphase. Considering the significance of Rho-mediated acto-myosin contractility at the onset of mitosis[25], we assume that Par3-dependent RhoA activities in interphase and early mitosis jointly determine mitotic fidelity. However, we do not exclude

 11

that altered contractility following Par3 loss additionally impacts on mechanisms, such as local force coupling between non-mitotic and mitotic neighbors[49], or delamination of basal epidermal cells in interphase[6,50–52], thereby potentially further contributing to premature differentiation.

This study sought to clarify the significance of polarity proteins in adult tissue homeostasis. There is increasing evidence that, in unperturbed tissues, the mechanisms balancing self-renewal and differentiation may vary between developing and adult (homeostatic) epidermis[5,6,52–55], likely due to very different demands and signaling pathways utilized during rapid tissue growth versus maintenance of tissue function. Local changes in cortical tension have recently been implicated in the process of epidermal stratification[49], highlighting the relevance of mechanical regulation for early tissue growth. Our findings add important roles of mechanocontrol for safeguarding self-renewal and differentiation in the adult skin epithelium. Contrasting the established function of Par3 in spindle orientation in several developmental systems[4,12,56,57], our data indicates that in fully developed, adult skin, polarity proteins help maintain tissue homeostasis primarily by steering keratinocyte mechanics and thereby mitotic fidelity and genome integrity, rather than by sole orientation of mitotic spindles. Next to causing mitotic errors and DNA-damage triggered differentiation, however, disturbed cortical contractility could also indirectly elicit the observed spindle orientation defects, e.g. due to improper spindle anchorage at the cortex, without that the latter necessarily impinges on cell fate. Such view of cortical tension acting upstream of LGN-NuMA localization and/or spindle orientation would be consistent with earlier observations made in epidermal Serum Response Factor-deficient mice[58] and fly imaginal disc epithelial cells with suppressed actomyosin function[59].

The reduced RhoA activity upon Par3 inactivation was accompanied by increased expression of p190-RhoGAP, a Rho-specific GAP previously implicated in RhoA inhibition at adherens junctions[60,61], in neuronal polarity[62], and as antagonist of Rho-GEFs during cell division[63]. Though the exact links between Par3 and RhoA-mediated signaling require further investigation, these findings open the possibility that Par3 mediates actomyosin contraction through restricting p190-RhoGAP function to shape a mechanical force platform at epidermal cell–cell contacts.

The mechanisms driving premature keratinocyte differentiation unraveled here might also contribute to the progressive HFSC decline of Par3eKO mice we reported recently[12]. We do not rule out additional mechanisms underlying this stem cell loss, also because the DNA damage in these mice was apparent throughout the skin epithelium, rather than reflecting a HFSC-restricted condition (Fig. 1a–g). However, the finding intrigued us that γH2Ax-positive cells were enriched in stem/progenitor compartments known to contain frequently cycling cells, such as the sub-bulge/secondary hair germ, the isthmus and the junctional zone[64–66]. Given the growing evidence for cross-talk and high plasticity of different skin stem/progenitor cell populations[67–71], the depletion of progenitor pools due to ectopic DNA-damage and resulting differentiation may compromise, directly or indirectly, the long-term maintenance of quiescent HFSCs. It will be interesting to identify how niche-specific mechanical regulation contributes to mitotic fidelity and resulting genome integrity of different epithelia. Finally, given the plethora of effects that DNA damage can yield in both tissue and cancer stem cells[72], and the consistent stabilization of p53 we observed upon Par3 loss (this study), it will be interesting to dissect in the future if mechanisms similar to those identified here underlie Par3-dependent and/or aPKCλ-dependent skin tumorigenesis[30,73].

Collectively, this study unravels a polarity protein-mediated mechanism that governs tissue homeostasis through coupling mechanical forces with mitotic accuracy and genome integrity,

thereby ultimately counteracting premature differentiation. Our findings identify mammalian Par3 as a key integrator of Rho-driven keratinocyte mechanics to balance self-renewal and differentiation. The mechanisms revealed here provide molecular and cellular insights into how polarity proteins integrate biomechanical signaling and cell division processes that are fundamental to maintain multilayered tissues.

## Methods

**Mice.** Mice with epidermal Par3 deletion have been described previously (K14CreneoKI[+/wt];Par3[fl/fl]; Par3eKO)[12]. For this study, we crossed Par3eKO mice onto a C57BL/6 background. Both male and female mice were included, with the male/female ratio comparable in the control and test groups, and for randomization, mice of different genotypes were co-housed. Mice were housed and fed according to federal guidelines. Conductance of all animal experiments presented in this study complied to all relevant ethical regulations for animal testing and research. The study received ethical approval by the State Office of North Rhine-Westphalia (LANUV), Germany. All animal experiments were performed according to institutional guidelines and animal licenses by the LANUV. Primers used for genotyping are listed in Supplementary Table 3.

**Keratinocyte isolation and culture.** Primary mouse keratinocytes were isolated from epidermis of newborn control and Par3eKO mice. To separate epidermis from dermis, whole skins of P0–P3 mice were incubated in 5 mg/ml Dispase II (Sigma-Aldrich) diluted in DMEM/HAM's F12 medium with low $Ca^{2+}$ (50 μM) (Biochrom) supplemented with 10% FCS (chelated), penicillin (100 U/ml), streptomycin (100 μg/ml, Biochrom #A2212), adenine ($1.8 \times 10^{-4}$ M, SIGMA #A3159), L-glutamine (2 mM, Biochrom #K0282), hydrocortisone (0.5 μg/ml, Sigma #H4001), epidermal growth factor (10 ng/ml, Sigma #E9644), cholera enterotoxin ($10^{-10}$ M, Sigma #C-8052), insulin (5 μg/ml, Sigma #I1882), and ascorbic acid (0.05 mg/ml, Sigma #A4034) at 4 °C overnight (termed FAD medium). After incubating the epidermis for 20 min in TrypLE (Gibco) at room temperature, dissociated cells were collected and cultured in low $Ca^{2+}$ FAD medium on collagen-I-coated plates. Mixed litters were used to isolate control and Par3eKO cells. For cell culture experiments, cell–cell contact formation was induced by calcium switch, i.e. by supplementing the FAD medium with 1.8 mM $CaCl_2$ for the indicated time points.

**Live cell imaging.** Briefly, primary keratinocytes isolated from H2B-GFP epidermal Par3KO (K14CreneoKI[wt/+];Par3[fl/fl];H2B-GFP[74]) and control mice (Par3[fl/fl];H2B-GFP) were seeded in μ-Slides (eight-well ibiTreat polymer coverslip; IBIDI) and calcium-switched for 24 h. In experiments with restoration of contractility, medium containing DMSO (vehicle control) or 1 nM Calyculin A (CalA) or 0.5 μg/ml CN03 was added 1 h before imaging. Time-lapse microscopy was performed using a Leica® DMI 6000 equipped with a Pecon® PM2000 incubator and a PlanApo ×20 0.75 NA objective. Cells were maintained at 37 °C, 5% $CO_2$, and time-lapse images were captured every 5 or 10 min for 16 h. Subsequent analysis of the material was performed with LAS X software (Leica).

**Immunocytochemistry.** Cells grown on permanox, glass chamber slides, or Transwell filters were fixed in 4% paraformaldehyde or EtOH/acetone, permeabilized with 0.5% Triton-X (if PFA fixed) in PBS++ (0.2 mM $Mg^{2+}$ and $Ca^{2+}$) and blocked in 5% bovine serum albumin (BSA) in PBS++. Samples were incubated overnight in primary antibodies diluted in AB buffer (10 mM Tris, 150 mM NaCl, 0.1%BSA, 0.02% sodium azide), followed by washing and incubation in secondary antibodies for 1 h at room temperature. Finally, samples were mounted in Mowiol.

**Immunohistochemistry.** For immunofluorescence staining of tissues, paraffin sections were deparaffinized, and antigens were retrieved by boiling in Dako Antigen Retrieval Agent pH 9 (Dako) for 20 min. After PBS wash, blocking was performed for 1 h with 5% BSA in PBS/PBS++ or Dako Antibody Diluent at RT. Primary antibodies diluted in either AB buffer or Dako Antibody Diluent were applied O/N at 4 °C in a humidified chamber. AlexaFluor-conjugated secondary antibodies and DAPI (Invitrogen, Darmstadt, Germany) were applied in Dako Antibody Diluent and incubated for 1 h at RT before washing and mounting in Mowiol. For immunohistochemistry of phosphorylated MLC2 Ser19 (Cell Signaling Technologies, #3675), above protocol was followed, with 5% BSA/PBS++ used for blocking non-specific binding sites, and Dako Antibody Diluent used for diluting primary and secondary antibodies plus DAPI. A step-by-step protocol of this procedure has been deposited[75].

**Preparation of cell extracts, SDS–PAGE, and immunoblotting.** Briefly, total cell lysates were generated by boiling cells in crude lysis buffer (10 mM EDTA, 1% SDS), protein concentrations were determined via BCA assay (Pierce, Thermo Fisher Scientific Darmstadt, Germany), and SDS–PAGE (8–12% PAA) and immunoblotting was performed according to standard procedures[74]. For better visualization, horizontally cropped Western Blots are displayed, showing the

relevant molecular weight range. In case of vertically cropped blots (non-neighboring bands on same blot), these were not fused but instead clearly marked with white separation in between. All immunoblots that are displayed in main and supplementary figures are shown as uncropped blots in Supplementary Figs. 10–12.

**Analysis of mitotic duration and aberrant divisions.** Time-lapse microscopy videos were used to analyze the mode of division, and divisions were categorized into normal divisions, lagging strands, tripolar divisions, and division with incomplete cytokinesis based on chromatin visualization by GFP-tagged histone H2B and differential interference contrast (DIC). For quantification of rescue experiments with restored contractility aberrant divisions have been normalized to the total number of aberrant division among all conditions. Mitotic duration was analyzed using time-lapse microscopy videos and defined as the difference between nuclear envelope breakdown and nuclear envelope reassembly. The frame of mitotic entry was defined as the first frame with a nucleus showing early prophase characteristics, such as initiation of chromatin condensation. The frame of cytokinesis was defined as the first frame with visible membrane reestablishment using DIC. Data from three individual experiments was pooled and used to calculate the mean event duration.

**Interphase fluorescence in situ hybridization.** Cells were trypsinized, and hypotonic lysis was achieved by resuspension in 75 mM KCl for 15 min at 37 °C, followed by fixation in Carnoy's fixative (75% methanol: 25% acetic acid) twice for 20 min at room temperature before storing samples at −20 °C until further use. Genomic DNA and iFISH probes (Tlk2 11qE1/Aurka 2qH3 mouse probe, Leica) were denatured at 73 °C for 3 min, and hybridization was performed at 37 °C overnight. Cells were subsequently washed with 0.4x saline-sodium citrate (SSC) (0.05% Tween-20) for 2 min at 68 °C and washed with 2xSSC (0.05% Tween-20) at room temperature for 6 min. Samples were dehydrated in ethanol series (70-90-100%), air-dried and mounted with VectaShield (Vector Laboratories) containing DAPI.

**Microscopy.** Confocal images were acquired with a Spinning Disk microscope (PerkinElmer UltraVIEW, Nikon) using a PlanApo ×40 0.95 NA air objective, and Zeiss Meta 710 using a Plan-Apochromat ×63/1.4 NA oil, Plan-Neofluar ×20/0,8, Plan-Neofluar ×40/1.3 Oil DIC. Epifluorescence images were obtained with a Leica DMI6000 and the following objectives: PlanApo ×63, 1.4 NA; PlanApo ×20, 0.75 NA. Look-up table (LUT) ranges are indicated in the figure legends.

**PIV and calculation of strain rates.** PIV was performed on 8-bit images of DIC/H2B-GFP from time-lapse micrograph sequences (5 min frame interval) using PIVlab[20] in MATLAB. Images were pre-processed using the Contrast-limited adaptive histogram equalization (CLAHE) value of 40 px and high pass filter of 20 px. The size of the interrogation window was of 128 × 128 pixels followed by three steps with subsequent half of the size of the previous. A Gaussian 2 × 3 fit was used for subpixel accuracy. Image pairs (1st–2nd, 2nd–3rd, etc.) were cross-correlated to extract the local displacement in between the two frames. Vector outliers were filtered manually via the Vector validation tool from the Post-processing menu, and after the removal of outliers, missing vectors were replaced by interpolated data from adjacent time points. Strain rates of mitotic cells were determined with the simplifying assumption that epithelial sheets upon junction maturation change from highly motile fluid to a solid, glass-like, state[76]. This change in behavior allows for accurate displacement measurements of mitotic cells once the junctions are mature and interphase cells are static. The low motion state of the epithelial sheets was confirmed, and then the displacement of mitotic cells was measured. To extract data from these mitotic fields, the PIVlab draw tool was used to create square ROIs of 176 × 176 pixels. By using the Extracting Parameters from Area in the Extractions menu of PIVlab, the Area Mean Value for the Strain rate [1 per frame] parameter was obtained. The strain rate is given in PIVlab as $e = \partial u/\partial x - \partial v/\partial y$[77].

**Analysis of intrinsic mitotic spindle geometry.** Keratinocytes were cultivated on chamber slides and switched to 1.8 mM Ca$^{2+}$ to induce cell–cell contact formation for 48 h. After fixation cells were immunostained for pericentrin and alpha-tubulin, and DNA was visualized using DAPI. Cells in metaphase were selected and the two angles $\alpha$ and $\beta$ between the metaphase plate and the spindle poles were determined in FIJI[78,79] using pericentrin and α-tubulin staining. The coherency index $c$ was calculated as follows: $c = \alpha/\beta$ (with: $\alpha \leq \beta \leq 90°$). For spindle shape analysis, thresholds for alpha-tubulin signals were set in FIJI (via AutoThreshold) and spindles identified manually with the Wand Tool. Measurements were performed with the Analyze Particles plugin.

**Quantification of protein signals in immunoblot analyses.** The band intensity of non-saturated Western blot signals was determined using the FIJI[78,79] rectangular too or GelAnalyzer on digital tiff-files of scanned Western blots. Bar diagrams show the relative protein or phosphorylation signals after normalization to loading controls and subsequent normalization to the sum of the densitometric signal of paired samples as described by Degasperi et al.[80].

**Plasmids and transfections.** A full-length Par3 (Par3-FL) expression construct was generated using the Gateway Cloning technology (Thermo Fisher Scientific). Murine Par3 cDNA was recombined into the pcDNA5/FRT/TO expression vector (Thermo Fisher Scientific). For membrane-localized Par3 (Par3-CAAX), an hRas CAAX motif generated by PCR (5′-CTT CTA CTC CCC TAG GGG CTG CAT GAG C-3′, 5′-GCG GCC GCT CAG GAG AGC ACA CAC TTG CAG CTC ATG CAG CC-3′) was fused to the C-terminus of Par3 (5′-ACG GGA ACA TTC CTT TCC AC -3′, 5′-AGC CCC TAG GGG AGT AGA AGG GCC G -3′) using fusion PCR, thereby introducing restriction sites for HindIII (5′-AGC CCC TAG GGG AGT AGA AGG GCC G -3′) and NotI (5′- GCG GCC GCT CAG G -3′, see all primers also listed in Supplementary Table 3). To obtain the Par3-CAAX expression plasmid, the HindIII-digested and NotI-digested PCR product and pcDNA5-Par3 were ligated using T4 DNA ligase (NEB). For ectopic expression, Par3-FL, Par3-CAAX, and pcDNA6/V5-HisC (Thermo Fisher Scientific, as vector transfection control) were transiently transfected into confluent primary murine *Par3*KO keratinocytes using ViromerRED® (Lipocalyx) and 0.43 μg DNA/cm$^2$ according to the manufacturer's protocol.

**Inhibitor treatment.** To restore actomyosin contractility, cells were treated with 1 nM Calyculin A (CalA, Cell Signaling Technologies Cat. No. #9902) or with 0.5–5 μg/ml Rho activator II (Cytoskeleton, Cat. no. CN03) for 24 or 48 h. Treatment with Cdk1 inhibitors Purvalanol (LC labs Cat. no., P-7890, 10 μM) or RO3306 (Sigma Aldrich, Cat. no. SML0569-5MG, 15 μM) for 48 h was used to block mitosis.

**SiRNA-mediated knockdown of p53 and Par3.** To down-regulate p53, 100% confluent primary keratinocyte monolayers were transfected using ViromerBLUE® (Lipocalyx) with 100 nM of ON-TARGETplus SmartPool targeting murine p53 or a non-targeting control siRNA pool (sip53: L-040642-00-0005, siCtrl: D-001206-14-20, Dharmacon) following the manufacturer's protocol. Cells were lysed 52 h after transfection and were cultivated for 48 h in FAD medium containing 1.8 mM Ca$^{2+}$. Knockdown efficiency was validated by Western blot analysis using p53-specific antibodies. To down-regulate Par3, 100% confluent primary keratinocyte monolayers were transfected as described above with 100 nM of ON-TARGETplus SmartPool targeting murine Par3 (equimolar mixture of four Par3 targeting SmartPools: J-040036-05, J-040036-06, J-040036-07, J-040036-08, Dharmacon) or a non-targeting control siRNA pool following the manufacturer's protocol. Cells were fixed 52 h after transfection and were cultivated for 48 h in FAD medium containing 1.8 mM Ca$^{2+}$. Knockdown efficiency was validated by immunocytochemistry.

**G-LISA® RhoA Activation Assay.** To determine the RhoA activity in control and *Par3*KO keratinocytes, the RhoA G-LISA® Activation Assay Kit (BK124-S, Cytoskeleton Inc.) was used. For this, primary keratinocytes were grown for 48 h in FAD/HC medium prior to cell lysis. Protein concentration was determined using the Precision Red Advanced Protein Assay (Cytoskeleton Inc.), and the protein concentration in lysates was adjusted to yield 4.5 mg protein/ml before performing the G-LISA RhoA activity assay. Absorption at 490 nm was measured, and spectrometry values were normalized to total RhoA levels as determined by Western blot analysis.

**Quantification of pMLC2, ppMLC2, and pERM immunoreactivity.** A custom FIJI[78,79] macro was used to draw constant lines of 15 μm length and 50 μm width across pMLC2 junctional sites. The mean gray values were used for quantification. The fluorescence intensity of immunohistochemistry micrographs was quantified by calculating the corrected total cell intensity fluorescence (CTCF) from skin histological sections. Maximum projections were generated from tile scan images, and the integrated density was measured using a ROI that encompassed the whole epidermis in FIJI. Integrated densities were plotted after subtracting the corresponding background signal measured within the tissue-free area.

**Quantification of pERM immunoreactivity during mitosis.** A Cell Profiler[81] pipeline was used to calculate pERM immunoreactivity in mitotic cells. Briefly, nuclei segmentation was achieved by global thresholding via the DAPI staining and used as seeds for primary object detection. pERM signals were segmented via manual thresholding, identified as objects and related to DAPI seeds. Analysis was done automatically with supervision, and data were exported to a spreadsheet. Due to the nature of the signal only metaphase/anaphase cells were considered.

**Atomic force microscopy.** Force spectroscopy was performed using a JPK Nanowizard Life Science instrument (JPK) mounted on an inverted optical microscope (Axiovert 200; Zeiss) for sample observation[82]. Briefly, $5 \times 10^5$ keratinocytes were seeded in 3.5 mm diameter plates (TPP) pre-coated with collagen-I, and switched to 1.8 mM Ca$^{2+}$ to induce cell–cell contact formation for 48 h. For nano-indentation experiments, silicon nitride pyramidal cantilevers (MLCT, Bruker Daltonics) with a nominal spring constant of 0.03 N m$^{-1}$, or spherical silicon dioxide beads with a diameter of 3.5 μm glued onto tip-less silicon nitride cantilevers (NanoAndMore/ sQube, CP-PNPL-SiO-B-5) with a nominal spring constant of 0.08 N m$^{-1}$ were used. The cantilever speed was kept constant at $v = 1.5$ μm/s,

and the force set-point was 0.05 nN. Data analysis was performed with Atomic J[83] and data fitting was automated accordingly to the tip geometry[82]. For pyramidal tips data was fitted to the approach curve using the classical golden, contact estimator, classical L2 fit and the pyramidal model (half-angle 17.5° degrees). For spherical tips, the approach curve was fitted to the classical golden, contact estimator, classical L2 fit and the Sneddon sphere model (radius 3.5 µm). For both cases we assumed cell incompressibility (Poisson ratio = 0.5) and read-in the calibration parameters (cantilever spring constant and inverse optical level sensitivity) from the force-curve files. The indentation was limited to 5–10% of the sample thickness.

**Quantification of γH2Ax-positive cells in the skin**. For quantification of γH2AX-positive cells in the interfollicular epidermis, z-stack images of γH2AX and DAPI stained back-skin cross sections were used. γH2Ax-positive cells were counted per 500 cells (DAPI) in the IFE. For quantification of γH2Ax-positive cells in the HF z-stack images of γH2AX, K15 (or CD34) and DAPI co-stained back-skin sections were used. Based on bulge marker expression (K15 and CD34) and histological hallmarks the HFs were separated into four compartments: The infundibulum, above bulge (isthmus and junctional zone), bulge (K15high, CD34+), below bulge (secondary hair germ, bulb, dermal papilla). For each compartment, γH2Ax-positive cells per number of total cells (DAPI) were counted. Data from five individual animals per genotype were pooled to calculate the percentage of positive compartments. Total numbers of positive and negative compartments per genotype were used for statistical analysis using the Fisher's exact test.

**RNA isolation from primary keratinocytes and epidermis**. Keratinocytes were cultivated on 10 cm dishes and switched to 1.8 mM Ca²⁺ to induce cell–cell contact formation for 48 h. Cells were washed with PBS, lysed with 600 µl RLT buffer (Qiagen, supplemented with 10 µM β-mercaptoethanol), and lysates were vortexed. For RNA isolation from P100 epidermis, dermis, and epidermis were separated using 0.8%Trypsin/PBS. Epidermal cells were collected in FAD medium and filtered using a 0.45 µm cell strainer. Cells were washed with PBS, lysed in 600 µl RLT buffer and homogenized using a syringe. For RNA isolation RNAeasy kit (Qiagen) was used according to the manufacturer's protocol.

**qRT-PCR**. RNA was transcribed into cDNA using QuantiTect Reverse Transcription Kit® (Qiagen). Quantitative real-time PCR was performed using different TaqMan gene expression assays® (ThermoFisher Scientific) listed in Supplementary Table 2. Gene expression changes were calculated using the comparative CT method and normalized to HPRT before normalization to control cells or epidermal lysates.

**UV irradiation**. Confluent primary keratinocyte cultures were washed once with PBS, then covered with PBS and irradiated with a dose of 100 mJ/cm² UV-B (304 nm). After irradiation, PBS was removed and cells were incubated at standard culture conditions for the indicated time points before protein lysates were collected.

**Epithelial sheet contraction assay**. $5 \times 10^5$ keratinocytes were plated in triplicates on six-well culture plates. 24 h after seeding, confluent cultures were switched to high calcium (1.8 mM Ca²⁺) for 48 h. Cells were subsequently washed with PBS⁺⁺ and incubated with 2 ml Dispase II (2.4U/ml in FAD medium with 1.8 mM CaCl₂) (Roche) per well, for 30 min at 37 °C, to detach the keratinocyte sheets. Remaining contacts at the sides of the wells were removed using a scalpel. Pharmacologic compounds were pre-incubated for 24 h prior to the assay and were present during dispase-mediated lift. Images of resulting contracted keratinocyte sheets were acquired using a GelDOC XR + (BioRad) station, and analyzed in FIJI[78,79].

**Stratified primary murine keratinocyte cultures**. $0.8–1.2 \times 10^5$ primary keratinocytes were seeded onto porous, collagen-I-coated TransWell filters (Corning, #3470, 0.4 µm pore size). 24 h after seeding confluent cultures were switched to high calcium (1.8 mM Ca²⁺) and subsequent stratification was allowed during a 10 days culture period. In case of inhibitor treatment, compounds were added at the time of calcium switch and replaced every second day.

**Slot blot**. Primary keratinocytes were cultured and UV-B irradiated using 3 mJ/cm² UV-B as described before. Cells were lysed using lysis buffer (0.2% SDS, 200 mM NaCl, 5 mM EDTA, 100 mM Tris–HCl, pH 8.5) and genomic DNA was extracted by isopropanol extraction. DNA was denatured for 5 min at 95 °C, put directly on ice, and blotted onto Amersham Hybond-N + membrane (RPN119B GE Healthcare) using a Whatman 96-well slot blotting device at 300 mbar vacuum. Cross-linking of the DNA was carried out for 2 h at 80 °C. The membrane was blocked for 30 min in 3% milk-PBS-T, incubated with anti-CPD antibodies (TDM-2, Cosmo Bio 1:10.000) overnight at 4 °C, and peroxidase-conjugated secondary antibodies (Jackson Immuno Research 1:10.000) for 1 h. After incubation with ECL (RPN2232, GE Healthcare), DNA Lesions were visualized and quantified on the Molecular Imager Gel Doc Imaging System (Bio-Rad) using Image Lab 5.0.

**Antibodies**. Detailed information on the antibodies used in this study is provided in Supplementary Table 1.

**Dephosphorylation of epitopes in tissue sections**. To validate the phospho-specificity of the pMLC Ser19 antibody used for immunohistochemistry of back-skin cross sections (Cell Signaling Technologies, #3675), sections were incubated with 25 U Antarctic phosphatase (AP) (NEB, M0289S) in AP buffer for 2 h at 37 °C. As negative control for the dephosphorylation, additional sections were incubated with AP buffer only and otherwise treated similar. Sections were then subjected to the standard pMLC2 immunohistochemistry protocol. As positive control for AP-mediated removal of phosphorylated epitopes, immunohistochemistry for γH2Ax was performed on consecutive sections. A step-by-step protocol of this procedure has been deposited[84].

**Rho activation in skin explants**. To increase contractility of cells within whole skin and to verify the specificity of the pMLC antibody (Cell Signaling Technologies, #3675) in immunofluorescence staining of back skin sections, 1 cm² back skin of control mice was excised and incubated in FAD medium (1.8 mM Ca²⁺) either containing DMSO or 5 µg/ml CN03 for 4 h at 32 °C. The tissue was fixed with 4% PFA for 1 h at room temperature and sections and immunohistochemistry were performed as described above.

**Statistical analyses**. Statistical analyses were performed using GraphPad Prism software (GraphPad, version 6.0). Significance was determined by Mann–Whitney $U$-test, Fisher's exact test, Student's $t$-test, Kruskal–Wallis ANOVA with Dunn's post-hoc test, One-Way ANOVA and Two-Way ANOVA with Tukey's multiple comparison or Sidak's multiple comparisons test, respectively, as indicated in the figure legends. All data sets were subjected to normality tests (D'Agostino–Pearson omnibus test, KS normality test, or Shapiro–Wilk normality test) when applicable. For statistical analysis of mitotic duration a mixed effects model was used in RStudio. $N$-values correspond to the sample size used to derive statistics. $p$-values are ranged as follows: $*p \leq 0.05$; $**p \leq 0.01$; $***p \leq 0.001$ as detailed in the figure legends. The number and type of biological replicates is specified in the figure legends. For all experiments shown in this study, measurements were taken from distinct samples, except for time-lapse microscopy (same samples were repeatedly imaged over time) and for atomic force microscopy (multiple subsequent indentations per sample).

**Software**. Data collection utilized the following software: Microscopy: LASX (Leica), ZEN (Zeiss), Volocity (PerkinElmer); Immunoblot: Samsung SCX3405 printer/scanner; GelDoc (Biorad); Atomic Force spectroscopy: JPK nanowizard; qRT-PCR: QuantStudio 12K Flex Software (Thermo). For data analysis, the following software has been used: GraphPad PRISM VI, RStudio, ImageJ/Fiji[78,79], Photoshop, MATLAB, PIVlab, CellProfiler, AtomicJ, GelAnalyzer 2010 and Excel 2010.

**Reporting summary**. Further information on research design is available in the Nature Research Reporting Summary linked to this article.

## Data availability

Correspondence and requests for materials related to this study should be sent to sandra.iden@uk-koeln.de. All data supporting the findings of this study are available within the paper and its supplementary information and source data files, or from the corresponding author on reasonable request.

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

## Acknowledgements

We thank Sina Knapp and Katherine Dodel for technical assistance, and the CECAD imaging facility and the UoC animal facilities for important services. We thank the Iden, Niessen, Wickström, and Bazzi laboratories as well as Christian Reinhardt and Jan Hoeijmakers for helpful discussions. Furthermore, we are grateful to Sara Wickström for providing tissues for antibody validation, to Floris Foijer for advice on iFISH, and to Mirka Uhlirova for various discussions and critical reading of the manuscript. We acknowledge Walter Birchmeier and Shigeo Ohno for sharing mouse lines. This project was funded by the Deutsche Forschungsgemeinschaft (DFG, German Research Foundation) (grants SPP1782-ID79/2-1, SPP1782-ID79/2-2, and Projektnummer 73111208-SFB 829, A10). Work in the Iden laboratory is further supported by the Excellence Initiative of the German federal and state governments (CECAD) and Center for Molecular Medicine Cologne (CMMC).

## Author contributions

Conceptualization, methodology, and validation: M.D.G., S.L., S.I.; investigation: S.L., M.D.G., M.S., S.I.; formal analysis: M.D.G., S.L., M.S., S.I.; resources: S.I.; visualization: M.D.G., S.L.; writing (original draft): S.L., M.D.G.; writing (review and editing): S.I.; supervision and funding acquisition: S.I.

## Additional information

**Competing interests:** The authors declare no competing interests.

