## [Peer Review File · Nature Communications]

When text is deleted in rebuttals and referee reports, add “[redacted]” in that location.

Reviewers' comments:

Reviewer #1 (Remarks to the Author):

Dias Gomes et al investigated a role of the polarity protein Par3 in epithelial homeostasis, which balances cell proliferation and differentiation. The authors showed that loss of Par3 in murine keratinocytes caused the activation of DNA damage response (upregulation of p53, pATR and pChk1), significant increase in aberrant cell divisions, and ectopic expression of Keratin1. They also demonstrated that Par3KO keratinocytes revealed a reduction in actomyosin contractility (lower levels of pMLC2, ppMLC2, and pERM). Notably, their data suggest that artificial increase in actomyosin contractility by treatment with 1 nM Calyculin-A restored the DNA damage response, aberrant cell divisions, and ectopic Keratin1 expression in Par3KO keratinocytes. Based on these findings, the authors proposed that Par3 acts to facilitate actomyosin contractility, which in turn prevents erroneous cell divisions and the DNA damage response.

This appears to be a well-executed study that highlights new function of Par3 in ensuring cell mechanics, which facilitates proper epithelial homeostasis. Although this paper does not provide molecular insights into how Par3 facilitates actomyosin contractility via phosphorylation of MLC2 and ERM, their results that suppression of Par3KO by artificial increase in actomyosin contractility provides substantial advance in our understanding of epithelial homeostasis and may contribute to further understanding of the maintenance of hair follicle stem cells. Therefore, in my opinion, these findings make this manuscript worthy for consideration in Nature Communications. Nevertheless, while the manuscript describes exceptionally important results in support of most of their conclusions, the authors first need to address the following issues before the manuscript can be considered suitable for publication.

Major point

I have a fundamental concern on their interpretation of Figs. 4L, 4M, 5A, and 5B, which are essential to support the main conclusion of this paper. Treatment with 1 nM Calyculin-A reduced the level of p53 and Keratin1 not only in Par3KO cells but also in wild-type cells. The authors mentioned in the main text that these reduction in wild-type cells are “non-significant”, but this claim was not validated by statistical comparisons in Figs. 4M and 5B. This evidence clearly indicates that Calyculin-A treatment can modify the DNA damage response and Keratin1 expression in a manner

independent of Par3. Therefore, it is hard to insist that Calyculin-A treatment was used as an activation of the Par3 effector pathway, thereby significantly weakening the author's claim that "Par3 acts upstream of Rho/ROCK /myosin to prevent erroneous cell divisions". Given that Calyculin-A is an inhibitor of number of phosphatases that function in multiple aspects of mitosis, I strongly suggest the authors to utilize other types of manipulation that specifically increases actomyosin contractility, such as a small peptide that stabilize RhoA-GTP or expression of constitutive active RhoA, to further validate if artificial increase in actomyosin contractility could suppress Par3KO phenotypes.

The same type of concern is also applied to Fig. 2F and 2G, in which treatment with PurvA reduced the level of pATR not only in Par3KO cells but also in wild-type cells.

Minor point

1. The authors repeatedly mentioned in the main text that "Par3 acts upstream of Rho/ROCK/myosin". However, there is no direct evidence that Par3 indeed activates the RhoA-ROCK signaling in this manuscript. The authors should provide more compelling evidence to support this claim; otherwise they need to weaken this description.

2. There is no direct evidence that supports a role of Par3 at cell-cell junctions in this manuscript, as they used Par3KO cells rather than specific inhibition of Par3 at cell-cell junctions.

3. Several claims in this manuscript are not statistically supported (Figs. 3B, 3C, 3G and 3H). In addition, the authors did not indicate the experimental replicates in several figure legends (Figs. 3G, 3H, and 4A).

4. I am wondering how the authors chose the two-tailed student t-test for the statistical tests which rely on smaller sample numbers (around or less than 5) in Figs. 1D, 1F, 2B (the statistical test used was not specified), and 3F.

5. A method section does not include in-detail procedures for calculating the strain rate from PIV images.

Reviewer #2 (Remarks to the Author):

This paper examines the source of increased DNA damage in mouse skin lacking the polarity protein Par-3. The authors convincingly trace down the origin of the defect to “aberrant” cell divisions and shed some light onto the nature of the aberrations by demonstrating that Par-3 is required for proper myosin 2 activation (light chain phosphorylation) and that a phosphatase inhibitor can rescue the defect. Overall this work represents a solid advance in our understanding of epithelial biology and I recommend publication once the following issues are addressed.

A key aspect of the model is that Par-3 regulates “actomyosin contractility to focus mitotic spindle” but I don’t feel the paper went far enough in explain how that might work (i.e. how Par-3 directed contractility regulation affects spindle focusing). Is Par-3 regulating myosin 2 light chain phosphorylation in a polarized manner? If so, can this be seen (i.e. does Par-3 colocalize with regions of increase pMLC signal in dividing cells?).

Personally, I found that the paper is somewhat confusing because it’s written in “mystery” format that seems to be according to the manner the authors made their discoveries. I hope they will consider an alternate format that begins with the most significant observation (that Par-3 is required for proper mitosis) and proceeds to explain both the mechanism and functional outcome (e.g. DNA damage). Since the “genomic integrity” aspect of Par-3 function is indirect, it doesn’t make sense to me to focus on it so much.

I assume the different symbols in 2c are meant to represent the different types of “aberrant” divisions, but they aren’t labeled.

Reviewer #3 (Remarks to the Author):

The paper from Dias Gomes, Letzian et al follows up on a prior study of the same group regarding the role of the polarity protein Par3 in epidermal differentiation and homeostasis. Their past study revealed epidermal specific knockout of Par3 gives rise to a barrier defect due to a role for polarity signalling upstream of tight junction assembly, as well as to a differentiation imbalance that leads to hyper-thickening. The current manuscript endeavours to provide mechanism for the latter phenotype and uncovers a role for Par3 in generating the cortical tension that is required for faithful mitotic control. In the absence of Par3, cultured keratinocytes show aberrant mitosis and aneuploidy, and they have reduced cortical actomyosin contractility. The authors suggest the tissue

mounts a DNA-damage response, which they show leads to P53 stabilization and increased differentiation of Par3-deficient cells in vitro.

Due to concerns about novelty and depth of insight as outlined below, the manuscript in its current form would be better suited for a more specialized journal than the audience of Nature Communications.

Major concerns:

1. While a few interesting observations are reported, the novelty of their conclusions are limited. The authors already describe many aspects of the Par3 skin phenotype in their prior paper (Ali et al, JID, 2016), and recent work from the Wickstrom group has highlighted the importance of balancing cortical tension for epidermal differentiation (Miroshnikova et al, NCB, 2018). It is also well established that cortical tension is required for effective control of mitosis (Lancaster et al, Dev Cell, 2013), and that defects in mitosis lead to premature keratinocyte differentiation (Shahbazi et al, JCS, 2017). One potentially distinct finding of their study is the implication that in the epidermis, Par3 is actually upstream of actomyosin contractility. Many other studies in flies and in MDCK suggest the converse – that polarized Par3 recruitment is directed by contractility. However, this finding unto itself is rather incremental.
2. The authors claim that increased DNA damage underlies the in vivo phenotype they observe. However, this element of the manuscript needs to be significantly improved. The staining of HFSCs is not clear and higher magnification and co-markers would be helpful. The choice of analysing quiescent HFSCs is not well justified, as the remainder of the manuscript focuses on actively cycling cultured keratinocytes. It would help to look downstream of p53 in vivo, particularly in epidermal cells, as well as in vitro.
3. The authors use multiple cell systems to reach their conclusions (cultured epidermal cells, in vivo bulge cells), and it is unclear how well their conclusions translate between these systems. This is important because mechanisms of cell division are highly dependent on the microenvironment. The mechanisms they uncover may be plausible based on their in vitro studies, but I am not sure how convinced I am that it accounts for the in vivo phenotype. Is there really enough DNA damage to drive the extent of hyper-thickening? More in vivo characterization should be included such as looking at H2AX and/or P53 in the IFE (they currently only show H2AX in the hair follicle). Their claim that pMLC2/pERM staining is altered in vivo is also not well supported by the low magnification immunofluorescence images they show; in fact, the pMLC2 looks like autofluorescence. Also, this time-point (P0) is far before the phenotype described in Ali et al manifests, so the relevance of looking at tissue at these time points is also questionable.

Minor concerns:

1. In Fig. 1a, Par3-KO it is not clear that the cells being highlighted are bulge stem cells (vs. inner bulge or hair germ). Better sections and higher magnification are needed to make this point.
2. Higher magnification images are needed in Fig. 3g,h
3. What is the calcium concentration in their cell culture experiments? Are they conducting a calcium switch?
4. Is there any evidence of stem cell exhaustion in Par3-KO? This would appear to be the case based on their model (Fig. 6).
5. The figure call outs to the supplemental figures are incorrect.
6. The figure legends do not explain different colours/shapes of data points plot in Fig. 2c.
7. The schematic in Fig. 6 implicates Chk1, but the data in Fig. 1 and Supplemental Fig. 1 shows no change in Chk1 levels (and also quantification isn't provided).
8. New data is introduced in the discussion (P-cad/DDR1 data) – this should be described properly in the results or removed entirely.

Point-by-point response to reviewers

Reviewers' comments:

Reviewer #1 (Remarks to the Author):

Dias Gomes et al investigated a role of the polarity protein Par3 in epithelial homeostasis, which balances cell proliferation and differentiation. The authors showed that loss of Par3 in murine keratinocytes caused the activation of DNA damage response (up-regulation of p53, pATR and pChk1), significant increase in aberrant cell divisions, and ectopic expression of Keratin1. They also demonstrated that Par3KO keratinocytes revealed a reduction in actomyosin contractility (lower levels of pMLC2, ppMLC2, and pERM). Notably, their data suggest that artificial increase in actomyosin contractility by treatment with 1 nM Calyculin-A restored the DNA damage response, aberrant cell divisions, and ectopic Keratin1 expression in Par3KO keratinocytes. Based on these findings, the authors proposed that Par3 acts to facilitate actomyosin contractility, which in turn prevents erroneous cell divisions and the DNA damage response.

This appears to be a well-executed study that highlights new function of Par3 in ensuring cell mechanics, which facilitates proper epithelial homeostasis. Although this paper does not provide molecular insights into how Par3 facilitates actomyosin contractility via phosphorylation of MLC2 and ERM, their results that suppression of Par3KO by artificial increase in actomyosin contractility provides substantial advance in our understanding of epithelial homeostasis and may contribute to further understanding of the maintenance of hair follicle stem cells. Therefore, in my opinion, these findings make this manuscript worthy for consideration in Nature Communications. Nevertheless, while the manuscript describes exceptionally important results in support of most of their conclusions, the authors first need to address the following issues before the manuscript can be considered suitable for publication.

We would like to thank the reviewer for appreciating the significance of our work and for his/her constructive and very useful comments that have helped us improve the manuscript considerably. Please find a point-by-point answer below.

Major point

I have a fundamental concern on their interpretation of Figs. 4L, 4M, 5A, and 5B, which are essential to support the main conclusion of this paper. Treatment with 1 nM Calyculin-A reduced the level of p53 and Keratin1 not only in Par3KO cells but also in wild-type cells. The authors mentioned in the main text that these reduction in wild-type cells are “non-significant”, but this claim was not validated by statistical comparisons in Figs. 4M and 5B. This evidence clearly indicates that Calyculin-A treatment can modify the DNA damage response and Keratin1 expression in a manner independent of Par3. Therefore, it is hard to insist that Calyculin-A treatment was used as an activation of the Par3 effector pathway, thereby significantly weakening the author’s claim that “Par3 acts upstream of Rho/ROCK /myosin to prevent erroneous cell divisions”. Given that Calyculin-A is an inhibitor of number of phosphatases that function in multiple aspects of mitosis, I strongly suggest the authors to utilize other types of manipulation

that specifically increases actomyosin contractility, such as a small peptide that stabilize RhoA-GTP or expression of constitutive active RhoA, to further validate if artificial increase in actomyosin contractility could suppress Par3KO phenotypes.

We thank the reviewer for raising this important aspect. Initially, we were also surprised that to some extent Calyculin A affects control cells. First, we would like to mention that we have carefully titrated the concentration of Calyculin A for our system. We only used a low dose (1nM) of this inhibitor in all experiments shown in the original and revised manuscript, thereby at least minimizing the risk of potential non-specific effects. Nevertheless, we agree with the concern raised, and therefore used additional means to restore contractility. Following this reviewer's suggestion, we restored Rho activity employing a Rho-activating peptide (CN03, Cytoskeleton) to examine its impact on intrinsic force generation, mitosis, and DNA damage responses. These studies revealed similar effects of low-dose Calyculin A and CN03 on *Par3KO* keratinocytes:

1. CN03 treatment was able to restore the contraction of Dispase-lifted *Par3KO* keratinocyte sheets to the level of control cell sheets (Figure 5d);
2. Time-lapse microscopy of primary *Par3KO* keratinocytes showed that CN03 treatment was sufficient to restore the mitotic fidelity of Par3-deficient keratinocytes to the level of control cells (revised Figure 6g), and
3. Western Blot analysis demonstrated reduced p53 levels of CN03-treated *Par3KO* cells upon restoration of Rho activity (revised Figure 6h,i).

Together, these new analyses thus confirmed our previous observations using low-dose Calyculin A. Notably, however, CN03 treatment, similar to Calyculin A treatment, also led to a reduction of p53 levels in control keratinocytes (compare revised Figures 6e,f,h,i). This data suggests that increasing actomyosin contractility *per se* -rather than non-specific effects of Calyculin A- to some extent already improves genome stability in a Par3-independent manner. The laboratory of Angelika Amon recently demonstrated that disrupting tissue architecture by taking primary cells into 2D culture could cause an increase in mitotic infidelity in a manner dependent on integrins (Knouse et al. Cell 2018, PMID 30146160). In line with that report, we observed that mitotic blockade slightly reduced the baseline DNA damage of control cells (see also comment 2 of this reviewer, and our original figure 2g plus note in the original manuscript). Further congruent with the Amon study, *in vivo* we did not detect marked signs of genome instability in adult skin of control mice, whereas Par3-deficient epidermis exhibited the described elevated DNA damage. Thus, we conclude that

- a) in line with the Knouse et al. paper, primary keratinocytes, compared to epidermal keratinocytes *in vivo*, show a baseline, Par3-independent mitotic infidelity and genome instability,
- b) Par3 inactivation leads to significantly reduced actomyosin contractility and a significant increase of mitotic infidelity and genome instability when compared to control cells,
- c) the baseline genome instability in control cells as well as the prominent deterioration due to Par3 loss can be suppressed by increasing actomyosin contractility (by use of Calyculin A or CN03), suggesting that Par3 is a major though not the exclusive upstream mediator of Rho/myosin activity in cultured keratinocytes.

Collectively, these data suggest that keratinocytes use both Par3-dependent and -independent pathways to regulate actomyosin contractility for faithful mitosis. We have now added this point in the revised manuscript.

Concerning the statistical analysis in figure 4M, 5B: Both the data normalization and the statistical information related to original figures 4M (revised figure 6e,f) and 5B (revised Figure 7a,b) have been revised. P-value ranges for all comparisons are shown in the revised figures, with the exact values stated in the corresponding legends.

The same type of concern is also applied to Fig. 2F and 2G, in which treatment with PurvA reduced the level of pATR not only in Par3KO cells but also in wild-type cells.

Concerning the (non-significant) effect of PurvalanolA on control cells (revised Figure 2g), we refer this reviewer also to our response to the previous, related comment. In line with the Knouse et al. paper (2018), we attribute the subtle improvement of pATR levels in mitotically blocked control cells to the suppression of DNA damage resulting from baseline, Par3-independent mitotic inaccuracy elicited by culturing cells *in vitro*.

Next to PurvalanolA, we have now also used the Cdk1 inhibitor RO3306 to block mitosis. In line with our previous findings using PurvalanolA, RO3306 treatment was able to suppress the ectopic activation of ATR (pATR) in *Par3KO* keratinocytes, yielding pATR levels comparable to control cells (new Suppl. Figure 2a,b). These results therefore confirm the significance of mitosis in causing DNA damage responses following Par3 loss. For reasons currently not known, at least qualitatively the pATR signals in control cells were less sensitive to RO3306 than to PurvalanolA treatment. Of note though, none of the inhibitors did elicit statistically significant differences in control cells (revised Figure 2f,g; Suppl. Figure 2a,b).

We have revisited the statistics and added a remark on this aspect in the revised manuscript (see revised figures, figure legends and results).

Minor point

1. The authors repeatedly mentioned in the main text that “Par3 acts upstream of Rho/ROCK/myosin”. However, there is no direct evidence that Par3 indeed activates the RhoA-ROCK signaling in this manuscript. The authors should provide more compelling evidence to support this claim; otherwise, they need to weaken this description.

We agree with the reviewer that the original data did not sufficiently support this statement. To provide direct evidence that Par3 acts upstream of Rho/myosin we now present results from a series of new experiments:

1. Rho activity assays: Using the RhoA G-LISA assay (Cytoskeleton) we now demonstrate that Par3 inactivation in primary keratinocytes results in significantly decreased levels of active RhoA (RhoA-GTP) but not of total RhoA (see revised Figure 3g, revised Suppl. Figure 5a,b);
2. Restoration of Rho-GTP levels in *Par3KO* keratinocytes using CN03, next to rescuing keratinocyte sheet contraction (revised Figure 5d), is able to restore mitotic fidelity (revised Figure 6g) and to reduce the levels of p53 (revised Figure 6h,i);

3. Ectopic expression of Par3-full length (FL) or of a membrane targeted version of Par3 (Par3-CAAX) is sufficient to significantly increase junctional pMLC2 in *Par3*KO cells (see also minor point 2) (revised Suppl. Figure 4a,b).
4. Though only correlative at this stage, we also found increased expression of the RhoA inhibitor p190-RhoGAP upon Par3 loss (revised Figure 3h,i). Future investigations outside the scope of this work will have to clarify if p190-RhoGAP is involved in Par3-mediated control of epidermal homeostasis.

Together with our findings reported in the original manuscript, these quantitative data on RhoA activity as well as the Par3- and RhoA *gain-of-function* approaches provide important new evidence that strongly supports our original hypothesis of Par3 acting upstream of Rho/myosin. We incorporated these new results in the revised manuscript (see revised results, discussion and figures).

As we did not directly activate ROCK itself, we have modified the text and removed it from the results, discussion and figures.

2. There is no direct evidence that supports a role of Par3 at cell-cell junctions in this manuscript, as they used Par3KO cells rather than specific inhibition of Par3 at cell-cell junctions.

We agree with the reviewer that genetic inactivation of Par3 alone is insufficient to formally show that junctional Par3 is the relevant pool regulating actomyosin contractility. This -admittedly simplified- assumption was made based on our observation that at those culture conditions (i.e. after calcium switch, allowing the establishment of cell-cell contacts), endogenous Par3 predominantly localizes to sites of cell-cell adhesion (Iden et al., Cancer Cell 2012, and revised Suppl. Figure 3a). Only at these conditions but not in the absence of cell-cell adhesions (i.e. low calcium conditions) Par3 was able to exert various signaling functions (Iden et al., Cancer Cell 2012).

At present, we unfortunately do not possess a validated system that enables us to inhibit Par3 specifically at cell-cell adhesions but not at other subcellular sites. To nevertheless more directly test whether junctional Par3 was required for actomyosin regulation in our system, we now generated a membrane-targeted version of Par3 (Par3-CAAX) and expressed this as well as Par3-full length (FL) or vector controls in primary *Par3*KO keratinocytes. Quantitative image analysis revealed that both Par3-FL and Par3-CAAX expression were sufficient to increase junctional pMLC2 when compared to vector-transfected *Par3*KO keratinocytes (revised Suppl. Figure 4a,b). These new data suggest that cortical Par3 is able to promote myosin activation.

Additionally, though not directly answering this point, we now also acutely inactivated Par3 in primary keratinocytes using siRNAs. Similar to our original *loss-of-function* data derived from knockout keratinocytes, and opposite to the Par3 *gain-of-function* data above, we observed a reduction in junctional pMLC2 after Par3 depletion (revised Suppl. Figure 3b-d).

3. Several claims in this manuscript are not statistically supported (Figs. 3B, 3C, 3G and 3H). In addition, the authors did not indicate the experimental replicates (Figs. 3G, 3H, and 4A).

We thank the reviewer for the note. Per the reviewer's suggestion, we have now added statistical information as well as mentioning of the number of experimental replicates.

Moreover, we performed a statistical analysis of the mean strain rate, revealing a significant reduction in *Par3*KO keratinocytes (additional graph, revised Figure 3b). Likewise, the revised micrographs for pMLC2 and pERM immunoreactivity *in vivo* are now supported by quantification and statistical analysis (revised Figure 4a-d).

4. I am wondering how the authors chose the two-tailed student t-test for the statistical tests which rely on smaller sample numbers (around or less than 5) in Figs. 1D, 1F, 2B (the statistical test used was not specified), and 3F.

Indeed, using a t-test on some of the original data sets was not optimal. We have now increased the number of biological replicates for several of the assays, followed by normality tests and revision of our initial normalization and statistical analyses. T-tests for comparison of two groups were used in those cases where the normality tests indicated a Gaussian distribution; otherwise, non-parametric tests were performed. We updated and expanded the corresponding information throughout all figure legends and the revised method section.

5. A method section does not include in-detail procedures for calculating the strain rate from PIV images.

We thank the reviewer for pointing this out. We have now added a detailed description of the method used for calculating the strain rate (see revised method section).

Reviewer #2 (Remarks to the Author):

This paper examines the source of increased DNA damage in mouse skin lacking the polarity protein Par-3. The authors convincingly trace down the origin of the defect to “aberrant” cell divisions and shed some light onto the nature of the aberrations by demonstrating that Par-3 is required for proper myosin 2 activation (light chain phosphorylation) and that a phosphatase inhibitor can rescue the defect. Overall this work represents a solid advance in our understanding of epithelial biology and I recommend publication once the following issues are addressed.

We thank the reviewer for appreciating the novelty of our findings, and thank him/her for the useful comments that helped improve the manuscript.

A key aspect of the model is that Par-3 regulates “actomyosin contractility to focus mitotic spindle” but I don’t feel the paper went far enough in explain how that might work (i.e. how Par-3 directed contractility regulation affects spindle focusing). Is Par-3 regulating myosin 2 light chain phosphorylation in a polarized manner? If so, can this be seen (i.e. does Par-3 colocalize with regions of increase pMLC signal in dividing cells?).

We have now investigated pMLC2 levels and localization *in vivo* and *in vitro*, as well as a potential colocalization of Par3 and pMLC2 in interphase and during different phases of mitosis. These analyses revealed:

- a) In primary keratinocytes Par3 indeed partially colocalizes with regions of elevated pMLC2 signal during interphase, mitotic rounding, and at the cleavage furrow in telophase/cytokinesis (revised Suppl. Figure 6a).
- b) In line with our finding in interphase cells (revised Figure 3c,d; revised Figure 4a,c; Suppl. Figure 3b,c), Par3 loss resulted in a decrease of pMLC2 throughout all phases of mitosis (revised Suppl. Figure 6a).
- c) We did, however, not observe a marked planar polarized distribution of cortical pMLC2, and hence no Par3-dependent alterations of such distribution.

The revised manuscript also contains a series of new data on reduced RhoA activity in *Par3*KO keratinocytes (revised Figure 3g; revised Suppl. Figure 5a,b), and different rescue approaches employing re-activation of Rho (revised Figure 5d, 6g-i) or expression of cortical Par3 (Par3-CAAX) in KO keratinocytes (revised Suppl. Figure 4a,b). This way, we further substantiated our initial hypothesis of Par3 acting upstream of Rho/myosin activation.

Previous and more recent reports have shown that increasing cortical contractility promotes spindle morphogenesis and architecture, and can help focus the mitotic spindle to yield a bipolar spindle geometry (e.g. Kunda et al. *Curr Biol* 2008, PMID 18207738; Lancaster et al., *Dev Cell* 2013, PMID: 23623611; Rhys et al., *JCB* 2018, PMID 29133484, and preprint by Matthews et al., *bioRxiv* 2019, <http://dx.doi.org/10.1101/571885>). Indeed, reconstituting actomyosin contractility in our system was sufficient to restore the intrinsic spindle geometry in *Par3*KO keratinocytes (revised Figure 6a-c). Thus, based on our combined results on viscoelastic properties, PIV/strain rate, spindle geometry analysis, mitotic fidelity, and the various rescue strategies, we suggest that Par3 at the cellular cortex is required to promote sufficient Rho/myosin-dependent contractility, thereby facilitating mitotic rounding to

help spatially restrict mitotic spindle components for successful execution of cell division.

Though at present not clear if of functional relevance, we also observed increased expression of the Rho-inactivating protein p190-RhoGAP in the absence of Par3 (revised Figure 3h,i). Future investigations outside the scope of this manuscript will be required to dissect the exact molecular networks connecting Par3, Rho, myosin, and different spindle components in dividing epidermal keratinocytes. We have revised our results and discussion on these aspects, also integrating further literature implicating cortical mechanics in faithful mitosis (see revised results and discussion on this).

Personally, I found that the paper is somewhat confusing because it's written in "mystery" format that seems to be according to the manner the authors made their discoveries. I hope they will consider an alternate format that begins with the most significant observation (that Par-3 is required for proper mitosis) and proceeds to explain both the mechanism and functional outcome (e.g. DNA damage). Since the "genomic integrity" aspect of Par-3 function is indirect, it doesn't make sense to me to focus on it so much.

We highly appreciate this reviewer's opinion on the article format. The proposed sequence would indeed work very well, too. After careful consideration, however, we decided to keep the original outline. This is because so far, the prevailing view in the skin field was that mitotic spindle orientation predominantly determines epidermal fate decisions (a concept mostly derived from developing epidermis). Our findings instead indicate that, in the context of homeostatic tissues, the control of keratinocyte mechanics downstream of Par3/Rho/myosin serves as alternative mechanism to spindle orientation in steering epidermal fate decisions. Considering the current concepts in the field, we felt that it was important to emphasize the tissue-level consequences of DNA damage and compromised genome integrity for epithelial homeostasis. We rephrased the manuscript text to better point out the significant finding of Par3 as a key regulator of mitosis and underlying mechanisms, and hope for this reviewer's understanding that the overall article outline stayed close to the original format.

I assume the different symbols in 2c are meant to represent the different types of "aberrant" divisions, but they aren't labeled.

The different symbols in original Figure 2c were referring to data points from three independent experiments. However, this way of presenting our data was not consistent throughout the manuscript. We realized this is confusing, and therefore adapted the figure to display all data points in uniform symbols per genotype (revised Figure 2c).

Reviewer #3 (Remarks to the Author):

The paper from Dias Gomes, Letzian et al follows up on a prior study of the same group regarding the role of the polarity protein Par3 in epidermal differentiation and homeostasis. Their past study revealed epidermal specific knockout of Par3 gives rise to a barrier defect due to a role for polarity signalling upstream of tight junction assembly, as well as to a differentiation imbalance that leads to hyper-thickening. The current manuscript endeavours to provide mechanism for the latter phenotype and uncovers a role for Par3 in generating the cortical tension that is required for faithful mitotic control. In the absence of Par3, cultured keratinocytes show aberrant mitosis and aneuploidy, and they have reduced cortical actomyosin contractility. The authors suggest the tissue mounts a DNA-damage response, which they show leads to P53 stabilization and increased differentiation of Par3-deficient cells in vitro

Due to concerns about novelty and depth of insight as outlined below, the manuscript in its current form would be better suited for a more specialized journal than the audience of Nature Communications.

Major concerns:

While a few interesting observations are reported, the novelty of their conclusions are limited. The authors already describe many aspects of the Par3 skin phenotype in their prior paper (Ali et al, JID, 2016), and recent work from the Wickstrom group has highlighted the importance of balancing cortical tension for epidermal differentiation (Miroshnikova et al, NCB, 2018). It is also well established that cortical tension is required for effective control of mitosis (Lancaster et al, Dev Cell, 2013), and that defects in mitosis lead to premature keratinocyte differentiation (Shahbazi et al, JCS, 2017). One potentially distinct finding of their study is the implication that in the epidermis, Par3 is actually upstream of actomyosin contractility. Many other studies in flies and in MDCK suggest the converse – that polarized Par3 recruitment is directed by contractility. However, this finding unto itself is rather incremental.

We thank the reviewer for providing constructive comments that helped improve our manuscript. Please find a point-by-point answer below.

We have cited these publications in the original manuscript because certain aspects of the studies were of relevance in the context of our findings. For example, the recent paper by Miroshnikova et al. 2018 represents an important study on epithelial development, reporting how the developing, undamaged epidermis coordinates stratification involving local changes in cortical tension. That study did not address how adult tissues balance self-renewal and differentiation, or how aberrant cortical tension is linked to mitotic fidelity and genome instability, and consequences for differentiation in a genomically stressed tissue. The Shahbazi et al. study in JCS, while very informative concerning CLASP2 loss-of-function phenotypes, has not gone beyond correlating mitosis defects in CLASP2-deficient cells with increased keratinocyte differentiation. We originally cited this paper as a study that *associated* DNA damage with keratinocyte differentiation.

We believe that our new data described in this manuscript builds on our previous published data and extends it and the cited publications by providing a mechanistic causal link between Par3 and mechanical signaling that is important to instruct mitotic accu-

racy and genome integrity, thereby safeguarding epithelial homeostasis in the adult skin. In summary, our revised manuscript shows that a) Par3 acts upstream of Rho/myosin-dependent control of cortical tension, that b) Par3 loss elicits mitotic infidelity and subsequent DNA damage, and that c) the resulting genome instability serves as selective pressure toward epidermal differentiation and suprabasal fate.

2. The authors claim that increased DNA damage underlies the in vivo phenotype they observe. However, this element of the manuscript needs to be significantly improved. The staining of HFSCs is not clear and higher magnification and co-markers would be helpful. The choice of analysing quiescent HFSCs is not well justified, as the remainder of the manuscript focuses on actively cycling cultured keratinocytes.

We thank the reviewer for pointing out this limitation in the initial analysis. In agreement with the reviewer (comment 3), we do not think that the DNA damage within the HFSC population alone accounts for the overall increased differentiation in multiple compartments of epidermal Par3-deficient skin. We also observed clear signs of DNA damage outside the bulge area. Though mentioned in the original manuscript, we did not provide quantitative support of this notion.

To substantiate our initial findings, we now performed a detailed analysis of γ H2Ax in the interfollicular epidermis and different hair follicle compartments of adult mouse skin. This also included initial staining optimization, use of additional markers, and acquisition of micrographs of improved quality (revised Figure 1a-g, revised Suppl. Figure 1a,b). Using co-immunostaining of γ H2Ax with Keratin15 or CD34, combined with histological hallmarks, we now quantified DNA damage-positive cells along the entire hair follicle, namely the infundibulum, the isthmus/junctional zone (“above bulge”), the bulge itself, and the secondary hair germ/bulb/dermal papilla (“below bulge”). This analysis revealed a marked increase of γ H2Ax in all HF compartments of *Par3eKO* mice (revised figure 1f,g). Together, our previous and these new results revealed that Par3 loss results in increased DNA damage responses not only in the hair follicle bulge but also throughout the pilosebaceous unit and the interfollicular epidermis, rather than reflecting a HF stem cell-restricted phenomenon. We added this important data and discussion in the revised manuscript.

Please also see our response to point 3 by this reviewer.

It would help to look downstream of p53 in vivo, particularly in epidermal cells, as well as in vitro.

We thank the reviewer for this suggestion, and have now performed qRT-PCR analyses to gain insight into potential p53 transcriptional targets altered following the loss of Par3 both *in vivo* (control and *Par3eKO* epidermis, P100) and *in vitro* (primary keratinocytes). Next to several commonly known p53-dependent transcriptional targets such as Bax and Noxa we also examined the expression of more recently identified targets that were selected based on transcriptome studies (e.g. Li et al. 2013, PMID 24003036) and a recent p53-centered screen (Janic et al. 2018, PMID 29892060). These analyses revealed that different “non-canonical” p53 targets were significantly altered following loss of Par3, including the epidermal differentiation marker TGM2 (upregulated), and Dishevelled1 (decreased), a key component of the Wnt pathway that confers stemness in skin and other different systems (see revised results section and Suppl. Figure 7e,f).

Beyond the suggested analysis of putative p53 downstream targets, we now also performed p53 *loss-of-function* experiments. In line with our previous hypothesis, siRNA-mediated depletion of p53 in *Par3*KO keratinocytes was able to suppress the ectopic differentiation of Par3-deficient keratinocytes (revised Suppl. Figure 7,c,d). The residual difference in Keratin1 between sip53 and *Par3*KO/sip53 cells is likely attributed to a remaining dosage effect, as *Par3*KO cells still retained relatively higher levels of p53 as compared to control cells using this knockdown approach (revised Suppl. Figure 7c). Together, these p53 *loss-of-function* analyses support our hypothesis that p53 mediates premature differentiation following Par3 loss.

3. The authors use multiple cell systems to reach their conclusions (cultured epidermal cells, in vivo bulge cells), and it is unclear how well their conclusions translate between these systems. This is important because mechanisms of cell division are highly dependent on the microenvironment. The mechanisms they uncover may be plausible based on their in vitro studies, but I am not sure how convinced I am that it accounts for the in vivo phenotype. Is there really enough DNA damage to drive the extent of hyper-thickening? More in vivo characterization should be included such as looking at H2AX and/or P53 in the IFE (they currently only show H2AX in the hair follicle).

- “*Is there really enough DNA damage to drive the extent of hyper-thickening? More in vivo characterization should be included such as looking at H2AX and/or P53 in the IFE (they currently only show H2AX in the hair follicle)*”:

We agree with the reviewer. We did not mean to explain any hyper-thickening of *Par3*eKO epidermis by increased DNA damage, as the epidermal thickness is only marginally increased in young mice and, in contrast to the HFSC decline and enhanced differentiation in various compartments, is not observed anymore with progressive age (Ali et al., 2016). We therefore do not suspect a causal relation between DNA damage following Par3 loss and hyper-thickening, and do not connect these two processes in the present manuscript.

Moreover, as stated by this reviewer, it seems rather unlikely that the observed differentiation phenotypes are solely caused by elevated DNA damage within the HFSC pool (we refer to our response to comment 2). In revised Figure 1a-g we present new data of statistically significant increase of γ H2Ax-positive cells in the entire hair follicle as well as in the interfollicular epidermis of *Par3*eKO mice, and discuss this data in the revised manuscript.

- “*... The authors use multiple cell systems to reach their conclusions (cultured epidermal cells, in vivo bulge cells), and it is unclear how well their conclusions translate between these systems. This is important because mechanisms of cell division are highly dependent on the microenvironment.*”:

The comment on the relevance of the microenvironment is well taken. As a common approach to assess mechanisms that may underlie *in vivo* phenotypes, we had decided to employ diverse *in vitro* systems to better understand and support our tissue-derived data. To stay as close as possible to effects exerted on epidermal keratinocytes *in vivo*, we performed our analyses with primary murine keratinocytes up to passage 2. Moreover, we included a filter-based stratification assay that to some degree can better mimic spatial features of the epidermis. Using this system, we were able to confirm not only the premature differentiation of Par3-deficient cells we reported *in vivo* (Ali

et al., 2016), but also the rescue thereof by reconstituting actomyosin contractility (revised figure 7c-e), similar to our data obtained from 2D cultures in this study. Nevertheless, as rightfully argued by this reviewer, these different *in vitro* culture conditions cannot fully recapitulate the *in vivo* microenvironment, a problem that many studies face. A recent publication by Angelika Amon and colleagues (Knouse et al. 2018, PMID 30146160) reported that disruption of tissue architecture, as occurring when cells are cultured in an artificial 2D condition, promotes chromosomal instability. In line with that report, we did not detect marked signs of genome instability in control adult skin *in vivo* (revised Figure 1a-g), but observed that mitotic blockade slightly reduced the baseline DNA damage of control keratinocytes *in vitro* (revised Figure 2f,g, Suppl. Figure 2a,b). This indeed suggests that *in vitro* culture conditions can augment mitotic infidelity when compared to the *in vivo* context. Importantly, however, both *in vivo* and *in vitro* Par3-deficient cells exhibit significantly increased DNA damage and epidermal differentiation when compared to controls, arguing against a strict dependency of the observed *Par3*KO phenotypes on culture-elicited environmental changes.

Further concordant with the comment of this reviewer on the dependency of cell divisions on the microenvironment, we note that in *Par3*eKO mice the extent of DNA damage differed between the various skin compartments, which each harbor distinct niches. It will be interesting to dissect in future work whether and how this depends on microenvironmental control of keratinocyte mechanics.

We have discussed above points in the revised manuscript (see revised results).

Their claim that pMLC2/pERM staining is altered in vivo is also not well supported by the low magnification immunofluorescence images they show; in fact, the pMLC2 looks like autofluorescence. Also, this time-point (P0) is far before the phenotype described in Ali et al manifests, so the relevance of looking at tissue at these time points is also questionable.

We agree with this reviewer that P0 is a less-relevant time point in this context. In the original manuscript, main Figure 3g,h showed pMLC2 and pERM immunohistochemistry data from adult tissues (P100), which is in line with Ali et al. 2016. Next to P100 skin, we had also provided P0 data in the Supplementary Information. Following the reviewer's comment, we now focused on relevant time points and expanded our analysis of pMLC2 and pERM at P100, including quantitative data. We tested various commercially available phospho-MLC2 antibodies in different fixation, blocking and epitope retrieval protocols, and used different secondary antibodies for detection. We further validated the mouse anti-pMLC2 (Ser19) antibody (Cell Signalling Technologies) used for the *in vivo* analysis of pMLC2 by costaining with survivin. The pMLC2 antibody clearly labelled distinct structures previously reported to be high in active myosin, such as the cleavage furrow in late telophase cells (see a separate .tif file in the manuscript tracking system visualizing this).

After validating the antibodies, we went on to examine immunohistochemistry samples of P100 skin cross-sections using a confocal microscope and 20x as well as 63x objectives for both overview and higher resolution images. These analyses revealed significantly reduced pMLC2 in the epidermis of *Par3*eKO mice (see revised Figure 4a,c) and a trend for reduced pERM levels in the absence of epidermal Par3 (see revised Figure 4b,d). We removed the former P0 data set from the supplementary information due to its low relevance for the phenotypes observed in adult mice.

Minor concerns:

1. In Fig. 1a, Par3-KO it is not clear that the cells being highlighted are bulge stem cells (vs. inner bulge or hair germ). Better sections and higher magnification are needed to make this point.

Per the reviewer's suggestion, we have performed additional experiments and included better images with the relevant markers (revised Figure 1a-g and Suppl. Figure 1a,b). Within the bulge compartment, we did observe γ H2Ax-positive cells in the basal layer of the bulge, though the predominant signal was found within the inner bulge (Figure 1c,d). As also detailed in our responses to comment 2 and 3, *Par3eKO* mice exhibit increased DNA damage and premature differentiation also in other, more proliferative compartments of the pilosebaceous unit (including the secondary hair germ) as well as in the epidermis. Therefore, we do not imply that this defect is restricted to the HFSC compartment (revised discussion).

2. Higher magnification images are needed in Fig. 3g,h

We have performed additional immunostaining and imaging, and now provide higher magnification to demonstrate pMLC2 (revised Figure 4a,c) and pERM (revised Figure 4b,d) signals in the epidermis and hair follicle.

3. What is the calcium concentration in their cell culture experiments? Are they conducting a calcium switch?

We have reviewed the experimental details in the relevant methods sections, and added these details where necessary. Primary keratinocytes were isolated and propagated in FAD medium containing 50 μ M extracellular calcium ions. For cell culture experiments, cells were then switched to medium containing millimolar concentrations of Ca²⁺ (1.8mM) to enable cell-cell contact formation.

4. Is there any evidence of stem cell exhaustion in Par3-KO? This would appear to be the case based on their model (Fig. 6).

Yes, we previously reported that epidermal deletion of Par3 results in a progressive decline of HFSCs, accompanied by an enrichment of committed progenitors and increased differentiation in the junctional zone, sebaceous gland, and interfollicular epidermis (Ali et al., 2016). In that study, we characterized the HFSC pools of *Par3eKO* mice in-depth using FACS and immunohistochemistry. These observations formed a key basis for the present study. Our new findings strongly suggest that a key mechanism through which Par3 safeguards epidermal homeostasis, and potentially stem/progenitor cell maintenance, is by controlling keratinocyte mechanics to ensure faithful mitosis, genome stability and balanced self-renewal.

5. The figure call outs to the supplemental figures are incorrect.

Figure call outs to supplementary figures have been reviewed and corrected.

6. The figure legends do not explain different colours/shapes of data points plot in Fig. 2c.

The different symbols in original figure 2c were referring to data points from three independent experiments per genotype. However, we also realized this may be confusing, and have now adapted the figure to display all data in unique symbols per genotype (revised Figure 2c).

7. The schematic in Fig. 6 implicates Chk1, but the data in Fig. 1 and Supplemental Fig. 1 shows no change in Chk1 levels (and also quantification isn't provided).

In the original manuscript we provided quantitative data that phospho-Ser345-Chk1, reflecting the activated Chk1, is significantly increased in UV-exposed *Par3*KO keratinocytes (original Figure 1e,f). We were unable to provide quantitative data on non-irradiated keratinocytes due to weak signals, which we attribute to the dynamic regulation of phosphorylated Chk1 and its a rather fast turnover (own unpublished observation in time-course experiments). Conclusively, in contrast to significantly increased pATR, γ H2Ax and p53, we could not provide sufficient experimental evidence of a strong Chk1 activation following Par3 loss, and therefore removed “pChk1” from the schematic in revised Figure 8a.

The data in Supplementary Figure 1d show wild-type keratinocytes and Cre-expressing keratinocytes (both Par3-proficient) to illustrate that K14-driven Cre expression itself does not elicit Chk1 activation in response to UV.

8. New data is introduced in the discussion (*P-cad/DDR1* data) – this should be described properly in the results or removed entirely.

As suggested, we have now removed this data, as at this point it does not advance the conclusions of the manuscript.

Reviewers' comments:

Reviewer #1 (Remarks to the Author):

The authors have extensively addressed the comments of all reviewers, and the manuscripts now includes an impressive set of new data (including treatments with CN03 Rho-activating peptide and RO3306 Cdk1 inhibitor) that further support the conclusions stated in the original submission. They have substantially revised the text to clarify their results and interpretations. In brief, the new data further supports their conclusion that Par3 is a major upstream regulator of the Rho pathway that facilitates proper cell divisions. The authors also clarified all concerns on their statistical tests and added in-detail methods to the revised manuscript. Again, although the mechanism by which Par3 activated the Rho pathway remains unclear, this paper reports the compelling data that links epithelial apical-basal polarity (Par3) to cortical tension-dependent control of epidermal cell division and differentiation, and thus represents a solid advance in our current knowledge of epithelial homeostasis. I would like to support publication of this manuscript in Nature Communications.

Fumio MOTEGI

Reviewer #2 (Remarks to the Author):

The authors have adequately addressed my comments and suggestions. In my opinion the revised manuscript is suitable for publication.

Reviewer #3 (Remarks to the Author):

The manuscript from Dias-Gomes, Letzian et al, is improved. The descriptive data regarding accumulation of DNA damage in the epidermis is better and the authors have strengthened their claim that Par3 functions upstream of actomyosin contractility. The writing of the paper is more cohesive and data presentation is more streamlined. The reviewer points were addressed to a

satisfactory degree and errors were corrected. Altogether, their data more convincingly support the model proposed.

That said, there are a few points that should be addressed:

1. The distinct contribution of this current study from that of Miroshnokova et al is understood and the reviewers raise some good points in their rebuttal letter. However, it merits better discussion in the manuscript text. Could the authors comment on the possibility of differential requirements for Par3-dependent control of contractility during mitosis versus during interphase? Although the scope of this paper principally focuses on mitosis, it would be pertinent to at least mention the possibility this mechanism may also be important for control of cellular mechanics during interphase, especially as cells exit the basal layer via extrusion. Moreover, it may be worthwhile to make mention of potential differences/similarities between development/growth versus homeostasis.

2. The immunofluorescence images of pMLC2 and pERM in control and Par3eKO skin are somewhat improved; based on what is outlined in the rebuttal letter, clearly a lot of effort was invested in these improvements. However, it is still confusing as to why even in the epidermal knock out, we don't see pMLC2 in the dermal compartment? The fact we don't see clearly see labelled fibroblasts really calls into question whether or not this antibody is working properly. Better images should be provided to convincingly support the authors' quantifications.

3. There is a mistake in the legend of Figure 1f – H2AX pos/neg labels are reversed in the legend of the right-most graph.

Point-by-point response to reviewer 3

Reviewer #3 (Remarks to the Author):

The manuscript from Dias-Gomes, Letzian et al, is improved. The descriptive data regarding accumulation of DNA damage in the epidermis is better and the authors have strengthened their claim that Par3 functions upstream of actomyosin contractility. The writing of the paper is more cohesive and data presentation is more streamlined. The reviewer points were addressed to a satisfactory degree and errors were corrected. Altogether, their data more convincingly support the model proposed.

We thank the reviewer for appreciating the improvements made during revision. Please find our point-by-point response below.

That said, there are a few points that should be addressed:

1. The distinct contribution of this current study from that of Miroshnokova et al is understood and the reviewers raise some good points in their rebuttal letter. However, it merits better discussion in the manuscript text. Could the authors comment on the possibility of differential requirements for Par3-dependent control of contractility during mitosis versus during interphase? Although the scope of this paper principally focuses on mitosis, it would be pertinent to at least mention the possibility this mechanism may also be important for control of cellular mechanics during interphase, especially as cells exit the basal layer via extrusion. Moreover, it may be worthwhile to make mention of potential differences/similarities between development/growth versus homeostasis.

Based on this reviewer's suggestion we expanded the in-text discussions on a potential role of Par3 in regulating cellular mechanics at interphase vs. mitosis and at development/growth vs. homeostasis, accompanied by additional references (see revised manuscript text).

2. The immunofluorescence images of pMLC2 and pERM in control and Par3eKO skin are somewhat improved; based on what is outlined in the rebuttal letter, clearly a lot of effort was invested in these improvements. However, it is still confusing as to why even in the epidermal knock out, we don't see pMLC2 in the dermal compartment? The fact we don't see clearly see labelled fibroblasts really calls into question whether or not this antibody is working properly. Better images should be provided to convincingly support the authors' quantifications.

We now provided **new representative micrographs for pMLC2** immunohistochemistry of control and *Par3eKO* skin, as indeed the previous examples with regard to the dermis were not optimal (see revised Fig. 4a). Of note, in *adult* skin we typically ob-

serve only moderate levels of pMLC2 in dermal fibroblasts. In addition, compared to the packed epidermis, the cellularity in the ECM-rich adult dermis is rather low. In *embryonic* skin, we typically detect higher pMLC2 signals in the -at this stage more abundant- dermal fibroblast, and lower pMLC2 intensity in the epidermis.

We also wish to emphasize that the immunohistochemistry (IHC) data on single-phospho-MLC2 presented in this manuscript is solely based on using the mouse anti-pMLC2 Ser19 antibody (CST #3675). Only with this single-phospho-MLC2 antibody we obtained consistent and reliable results *in vivo*. An alternative product (CST #3671, rabbit anti-pMLC2 Ser19) did yield rather variable results in IHC, in line with reports by other skin experts (personal communications).

To our knowledge, there are no publications yet showing pMLC2 immunohistochemistry in adult skin using the #3675 antibody. However, different laboratories have successfully used this antibody in other systems like the mouse heart, zebrafish embryo, *Drosophila* retina, human breast cancer, and mouse and human arteries. Below we append a **list of these exemplary publications** and references to the relevant figures showing mouse anti-pMLC2 Ser19 IHCs (CST #3675).

Moreover, we have now performed a series of **additional validation experiments** that demonstrate the specificity of the CST #3675 antibody in our IHC protocol:

- **Alkaline phosphatase treatment:** We show that the #3675 antibody recognizes a phosphorylated epitope, as incubation of adult skin cross sections with Antarctic Phosphatase (an alkaline phosphatase, AP) prevents the IHC signals we report in this manuscript and that were also obtained in sections incubated in AP buffer only (see new Suppl. Fig. 5d,e). As positive control, IHC for phosphorylated H2Ax (gH2Ax) in *Par3eKO* skin was performed in parallel, revealing a loss of the distinct, nuclear gH2Ax immunostaining after AP treatment (see new Suppl. Fig. 5c).

[Redacted]

- **Rho activation in skin explants:** To elevate endogenous dermal pMLC2 signals we incubated adult skin explants with the Rho activator CN03 (or vehicle) *ex vivo*, followed by IHC for pMLC2 (CST #3675). CN03 treatment led to an increase of the existing pMLC2 signals in different skin cell types including fibroblasts and keratinocytes (see new Suppl. Fig. 5a,b). This data thus supports that our IHC protocol using CST #3675 results in specific labelling of a phosphorylated epitope downstream of Rho activation in the skin.
- Additionally, the revised manuscript now includes micrographs that illustrate **pMLC2 immunostaining in other areas previously reported to be high in active myosin**. The pMLC2 antibody (CST #3675) clearly labelled diverse blood vessels present in the papillary and reticular dermis as well as hypodermis (with

particularly high signals in smooth muscle layer) (see new Suppl. Fig. 5g). Similarly, distinct epidermal structures such as the cleavage furrow in late telophase cells are recognized by CST #3675 (costaining with survivin; in first revision provided for reviewer only) (see new Suppl. Fig. 5f).

Finally, we now also added new data using the **rabbit anti-ppMLC2 Thr18/Ser19** antibody (CST #3674), demonstrating that loss of epidermal Par3 significantly reduces this double-phosphorylated pool of MLC2, too (see new Suppl. Fig. 5h). Compared to the single phospho-MLC2 antibody (CST #3675), this antibody also yields proper signals in the skin, however, with somewhat higher variability in between biological replicates (as also reflected in the quantification provided in new Suppl. Fig. 5i). For this reason, we prefer to use the #3675 product for IHC in adult tissues.

Collectively, the revised manuscript contains a diverse set of genetic, biochemical and histochemical validation data that together support that the reduced pMLC2 IHC signal we detect in adult *Par3eKO* mice is based on a specific IHC protocol using the CST #3675 pMLC2 antibody.

3. There is a mistake in the legend of Figure 1f – H2AX pos/neg labels are reversed in the legend of the right-most graph.

We thank the reviewer for picking this up. The error has been corrected.

Related to point 2:

Examples of recent publications using specifically mouse anti-pSer19-MLC2 antibody (CST #3675):

Publication	Tissue	Figure reference	Comment
Francou et al., Nature Communications 2017 https://www.nature.com/articles/ncomms14770 PMID 28357999	Mouse embryonic heart	Suppl. Fig. 4c,d	
Zuo et al., Breast Cancer Research 2018 https://doi.org/10.1186/s13058-018-0966-2 PMID 29769144	Human breast tumor	Fig. 4a	
Alégot et al., eLife 2018 https://doi.org/10.7554/eLife.32943 PMID 29420170	Drosophila follicular epithelium	Fig. 5a, Suppl. Fig. 3	
Zihni et al., Nature Cell Biology 2017 https://doi.org/10.1038/ncb3592 PMID 28825699	Drosophila photoreceptors	Fig. 2b	
Li et al., Medical Science Monitor 2016 https://doi.org/10.12659/MSM.900152 PMID 27643564	Mouse and human arteries	Fig. 2, 4, 5	
Marei et al., Nature Communications 2016 https://doi.org/10.1038/ncomms10664 PMID 26887924	Human fibroblast matrices (in vitro, 3D)	Suppl. Fig. 7c	no representative micrographs provided, only quantification
Raman et al., Nature Communications 2016 https://doi.org/10.1038/ncomms11643 PMID 27249668	Zebrafish embryo	Fig. 7f	either CST #3675 or 3671 was used, both listed in methods